# Diversity of platinum-sites at platinum/fullerene interface accelerates alkaline hydrogen evolution

Jiayi Chen[1,7], Mohammed Aliasgar[1,7], Fernando Buendia Zamudio[1], Tianyu Zhang[1], Yilin Zhao [1], Xu Lian[2], Lan Wen[1], Haozhou Yang[1], Wenping Sun [3] ✉, Sergey M. Kozlov [1] ✉, Wei Chen[2,4,5,6] & Lei Wang [1,6] ✉

Membrane-based alkaline water electrolyser is promising for cost-effective green hydrogen production. One of its key technological obstacles is the development of active catalyst-materials for alkaline hydrogen-evolution-reaction (HER). Here, we show that the activity of platinum towards alkaline HER can be significantly enhanced by anchoring platinum-clusters onto two-dimensional fullerene nanosheets. The unusually large lattice distance (~0.8 nm) of the fullerene nanosheets and the ultra-small size of the platinum-clusters (~2 nm) leads to strong confinement of platinum clusters accompanied by pronounced charge redistributions at the intimate platinum/fullerene interface. As a result, the platinum-fullerene composite exhibits 12 times higher intrinsic activity for alkaline HER than the state-of-the-art platinum/carbon black catalyst. Detailed kinetic and computational investigations revealed the origin of the enhanced activity to be the diverse binding properties of the platinum-sites at the interface of platinum/fullerene, which generates highly active sites for all elementary steps in alkaline HER, particularly the sluggish Volmer step. Furthermore, encouraging energy efficiency of 74% and stability were achieved for alkaline water electrolyser assembled using platinum-fullerene composite under industrially relevant testing conditions.

Anion exchange membrane (AEM) based water electrolyser has attracted tremendous research interest due to the rapid development of efficient and abundant anodic catalysts in alkaline electrolytes[1–8]. While the unsatisfactory durability and ion conductivity of the alkaline exchange membrane still hinder the practical implementation of AEM water electrolyser, there are promising candidates reported recently which may lead to breakthroughs for this technology[9,10]. Thus, it remains desirable to develop efficient alkaline HER catalyst, since the electricity cost (large overpotential required causing high energy demand) is recognized to dominate the overall green hydrogen production cost[9,11]. However, the kinetics of the cathodic hydrogen evolution reaction (HER) in alkaline media remain unsatisfactory for practical applications. Even for the state-of-the-art platinum (Pt) catalysts, the activity decreases by up to a few orders of magnitude when pH changes from acidic to alkaline[12–15], due to still debated reasons[16–20]. Nonetheless, a handful of strategies have been explored to promote

[1]Department of Chemical and Biomolecular Engineering, National University of Singapore, 4 Engineering Drive 4, Singapore, Singapore. [2]Department of Chemistry, National University of Singapore, 3 Science Drive 3, Singapore, Singapore. [3]School of Materials Science and Engineering, State Key Laboratory of Clean Energy Utilization, Zhejiang University, Hangzhou 310027, P. R. China. [4]Department of Physics, National University of Singapore, 2 Science Drive 3, Singapore, Singapore. [5]Joint School of National University of Singapore and Tianjin University, International Campus of Tianjin University, Binhai New City, Fuzhou, China. [6]Centre for Hydrogen Innovations, National University of Singapore, 1 Engineering Drive 3, Singapore, Singapore. [7]These authors contributed equally: Jiayi Chen, Mohammed Aliasgar. ✉e-mail: wenpingsun@zju.edu.cn; sergey.kozlov@nus.edu.sg; wanglei8@nus.edu.sg

the HER activity of Pt-based catalysts in alkaline media[21–25]. First, facet-dependent alkaline HER activity was demonstrated on the Pt surface[26]. At low overpotential regions, the intrinsic activity increases in the sequence of Pt(111) <Pt(100) <Pt(110) for alkaline HER, in line with the Brønsted-Evans-Polanyi principle. Inspired by this, various transition metals were employed to alloy with Pt to tune the adsorption strength of hydrogen/hydroxide, and some of them achieved improved HER activity[12,21,27–31]. Besides, Pt decoration by materials with strong oxygen affinities, e.g., Ru adatoms[8,17], Ni(OH)₂ clusters[27], has been shown to accelerate the reaction kinetics toward alkaline HER[8,17,21,23–27]. Also, alkali-metal cations can promote the HER kinetics in the order of Li⁺ <Na⁺ <K⁺. This trend has been rationalized by the strong interactions between the relatively weak hydrated cations and the Pt surface, which can optimize the interfacial water structure and promote the Volmer step by stabilizing its transition state[22,25,32–35]. Moreover, tuning the valence state and chemical environment of the atomically dispersed Pt on carbon and/or other substrates is capable of modulating the Pt−H/Pt−OH interactions, subsequently accelerating the alkaline HER rate. Finally, metal–support interactions have been recognized as an effective approach to modifying the electronic structure of the active sites enhancing HER activity[25,28,36–40]. While the above strategies have shown promise in accelerating alkaline HER on Pt, it remains a general challenge to improve the intrinsic activity of Pt, since typical catalyst modifications accelerate certain reaction steps at the cost of slowing down other steps due to the scaling relations. Thus, in this work we tackle this challenge by constructing Pt-sites with diverse binding properties towards different key reaction intermediates to achieve improved rates for every elementary step of HER and enhanced intrinsic activity. Moreover, we believe this strategy can be applied broadly to the design of other catalysts, including earth-abundant transition metal HER catalysts to further reduce the cost of AEM devices.

Buckminsterfullerene ($C_{60}$) surfaces stand out as promising catalyst-support candidates due to their high electron affinity (each $C_{60}$ molecule can accept up to six electrons)[41], uniform and undulated structures with large lattice distance. These desired surface properties could enable strong electronic and confinement effects when anchoring metal nanoclusters/particles, and further lead to diverse active sites as we proposed above. Moreover, the strong intermolecular interactions between $C_{60}$ molecules can enable the formation of $C_{60}$ crystals with thin and highly dispersible morphology (i.e., 2D nanosheets)[42,43], which further enable sufficient electron conductivity[44–47] and high catalyst-loading-capacity for application as efficient electrocatalyst support. With this design in mind, we developed a facile approach to synthesize two-dimensional $C_{60}$ nanosheets, and constructed a $C_{60}$/Pt heterostructure ($PtC_{60}$) as a model system to demonstrate the above catalyst design principle. As expected, pronounced charge redistributions occur at the diverse interface of Pt/$C_{60}$, which leads to remarkably enhanced intrinsic activity (12 times) for alkaline HER compared to the state-of-the-art Pt/C catalyst. With a modest loading of ~0.4 mg cm⁻² on the rotating disk electrode, $PtC_{60}$ possesses an overpotential as low as 24 mV at 10 mA cm⁻² for HER in 1.0 M KOH. Experimental characterizations and kinetic simulations based on density functional theory (DFT) revealed that the higher activity of $PtC_{60}$ catalysts is due to the diversity of binding properties of Pt-sites at the Pt/$C_{60}$ interface towards the key reaction intermediates (e.g., hydrogen, hydroxide and adsorbed water), some of these interface sites are highly active in the Volmer step, whereas others are highly active in water dissociation and Heyrovsky steps, so that all HER steps are accelerated on Pt/$C_{60}$. Furthermore, we demonstrate the high activity of $PtC_{60}$ in membraned-based AEM electrolyser, which exhibits high energy efficiency of ~74% and adequate stability without further optimizations of other electrolyser components, making $PtC_{60}$ a promising candidate for practical AEM electrolysis.

## The synthesis and morphological characterizations of $PtC_{60}$

The $PtC_{60}$ nanosheets was synthesized by first preparing the $C_{60}$ bulk crystals through a modified approach of liquid-liquid interfacial precipitation (Fig. 1a), and then a one-step exfoliation of the $C_{60}$ crystals induced by direct solution-deposition of Pt clusters[46,48]. To facilitate the exfoliation, we introduced ethanol instead of isopropanol as the upper solution (Fig. 1a), and a lower temperature (5 °C) instead of room temperature to reduce the crystal size of $C_{60}$. As shown in the transmission electron microscopy (TEM, Supplementary Fig. 1a, b), the obtained $C_{60}$ crystals exhibit sizes ranging from 200 nm to 1 μm, much smaller compared to $C_{60}$ crystals reported elsewhere[46,48–50]. The TEM and Fast Fourier transform (FFT, Supplementary Fig. 1c, d) images indicate that the $C_{60}$ crystals display high crystallinity along with the (1$\bar{1}$0) basal plane with markedly irregular edges and an average thickness of ~20 nm (Atomic Force Microscope (AFM) images in Supplementary Fig. 1e, f), which are beneficial for the subsequent exfoliation step. X-ray diffraction (XRD) patterns of the $C_{60}$ crystals show a typical face-centred-cubic (fcc) phase (ICSD-73661, a = 14.16 Å, Supplementary Fig. 2) in line with the TEM results. During the exfoliation, the formation of Pt-metal-clusters and hydrogen gas from the vigorous chemical reduction of $H_2PtCl_6$ precursor by $NaBH_4$ (Eq. (1)) can overcome the interlayer interactions of the $C_{60}$ bulk crystals along with the [1$\bar{1}$0] axis, resulting in desired thin layers of $PtC_{60}$ nanosheets (Fig. 1a).

$$H_2PtCl_6 \cdot 6H_2O + NaBH_4 = Pt + B(OH)_3 + H_2 + HCl + NaCl \qquad (1)$$

AFM (Fig. 1e, Supplementary Fig. 3) and TEM images (Supplementary Fig. 4) suggest that the size of $PtC_{60}$ nanosheets ranges from 200 to 1000 nm with a thickness of ~5 nm. The profound Tyndall effect (Supplementary Fig. 5) further confirms the thin nature of the $PtC_{60}$ nanosheets. The FT-IR spectrum of $PtC_{60}$ (Supplementary Fig. 6) exhibits weakened peaks at 1175 cm⁻¹ and 1430 cm⁻¹ compared with that of the $C_{60}$ precursor, suggesting enhanced intermolecular interactions among $C_{60}$ molecules within $PtC_{60}$[51,52]. The cubic-phased XRD patterns of $C_{60}$ remain identical after the Pt intercalations (Supplementary Fig. 2), suggesting no phase changes occur during the exfoliation. In addition, the uniquely large interplanar spacing of the (111) and (200) planes of $C_{60}$ nanosheets can be observed from the high-resolution TEM. The corresponding FFT, and inverse FFT images of the $PtC_{60}$ (Fig. 1b–d), confirm that the exfoliation of $C_{60}$ by Pt deposition occurs along with the [1$\bar{1}$0] axis. Note that similar Pt nanocluster sizes (~2 nm) were obtained for $PtC_{60}$ with different Pt loadings, and these Pt nanoclusters disperse uniformly on the $C_{60}$ nanosheets in an unusually dense manner without any noticeable aggregations (Fig. 1g, Supplementary Figs. 7, 8). This phenomenon indicates the existence of a strong confinement effect on the Pt nanoclusters from the $C_{60}$ plane, which likely resulted from the sub-nanoscale roughness of the $C_{60}$ surface. We further evaluate this confinement effect under elevated temperatures. Encouragingly, the anchored Pt-nanoclusters remain well dispersed even under 300 °C (Supplementary Fig. 9), and show stability comparable if not more stable compared to the commercially optimized Pt/C (Supplementary Fig. 10). In contrast, obvious aggregations occurred to Pt nanoparticles with or without XC-72 carbon black substrate (Supplementary Figs. 11, 12) after the calcination treatment, further confirming the strong confinement effect induced by the $C_{60}$ support. This remarkable microstructural stability of Pt clusters on the $C_{60}$ nanosheet makes $C_{60}$ nanosheet a promising candidate as substrate material for constructing electrocatalysts with long-term durability. Separately, lattice distances of 0.22 nm and 0.2 nm are observed for Pt nanoclusters on $C_{60}$ nanosheets (Fig. 1f, and Supplementary Fig. 13a, b), corresponding to the fcc (111) and (200) crystal planes, respectively. Moreover, both the FFT patterns (Supplementary Fig. 13c) and XRD patterns (ICSD-64924, Supplementary Fig. 2) suggest that Pt nanoclusters have an fcc structure similar to that

of the pristine Pt NCs. Overall, the morphological characterization of the PtC$_{60}$ illustrates that C$_{60}$ nanosheets are a promising candidate as catalyst substrates on which small Pt nanoclusters can be uniformly and firmly anchored.

## The electronic structure of PtC$_{60}$

The electronic structure and chemical environment of PtC$_{60}$, particularly at the Pt/C$_{60}$ interface, were investigated by X-ray photoelectron spectroscopy (XPS) and X-ray absorption spectroscopy (XAS). The XPS survey spectrum (Supplementary Fig. 14) confirms the existence of Pt, C, and O in PtC$_{60}$, with no measurable impurities. The high-resolution XPS Pt $4f$ signals for Pt NCs can be deconvoluted into two components, corresponding to metallic Pt$^0$ and oxidized Pt$^{2+}$ species, at 71.2/74.5 eV and 72.6/76.3 eV, respectively (Fig. 2a)[28,37,53]. However, the broader Pt $4f$ peaks of PtC$_{60}$ compared to those of the Pt NCs suggest an extra content of Pt$^{\delta+}$ species. To explore the origin of the increased Pt$^{\delta+}$ species, e.g., whether the C$_{60}$ substrate stabilizes the oxidized form of Pt, we conducted curve fittings for the associated O $1s$ species for both Pt NCs, PtC$_{60}$ and C$_{60}$ precursor powder (Supplementary Fig. 15). It turns out that we could not find an increase of O $1s$ that is associated to the Pt oxide. Thus, based on the computational results discussed later, we tentatively attribute the extra content of Pt$^{\delta+}$ species in PtC$_{60}$ to the interfacial electron transfer from Pt to C$_{60}$. Moreover, we have prepared additional control catalysts: commercial Pt/C (20% Pt loading) and Pt deposited on XC-72 carbon black (Pt/CB) (40% Pt loading)). The Pt $4f$ XPS of Pt/CB exhibits sharp peaks and peak compositions similar to those of the Pt NCs, suggesting that the electronic interactions between Pt and graphitic carbon are negligible (Supplementary Fig. 16). In the following, we will focus only on the analysis of Pt/C catalysts because they feature similar intrinsic activity and HER mechanism as Pt/CB according to the Tafel plots (Supplementary Fig. 17).

Based on work function of C$_{60}$ (work function of ~4.5–5 eV)[54] and Pt (work function of 5.12–5.93 eV)[55], one would expect that the electron transfer occurs in the opposite direction as we observed by XPS. Similar counterintuitive phenomenon was observed elsewhere, which has been rationalized by that the charge accumulation only occurs on a small portion of metal and carbon atoms at the interface[56–58]. To prove the presence of interfacial charge transfer, we conducted all-in-vacuum XPS and ultraviolet photoelectron spectroscopy (UPS) measurements by keeping the deposition of C$_{60}$ thin-film and Pt layer within the same vacuum chamber, (Supplementary Fig. 18) so that the

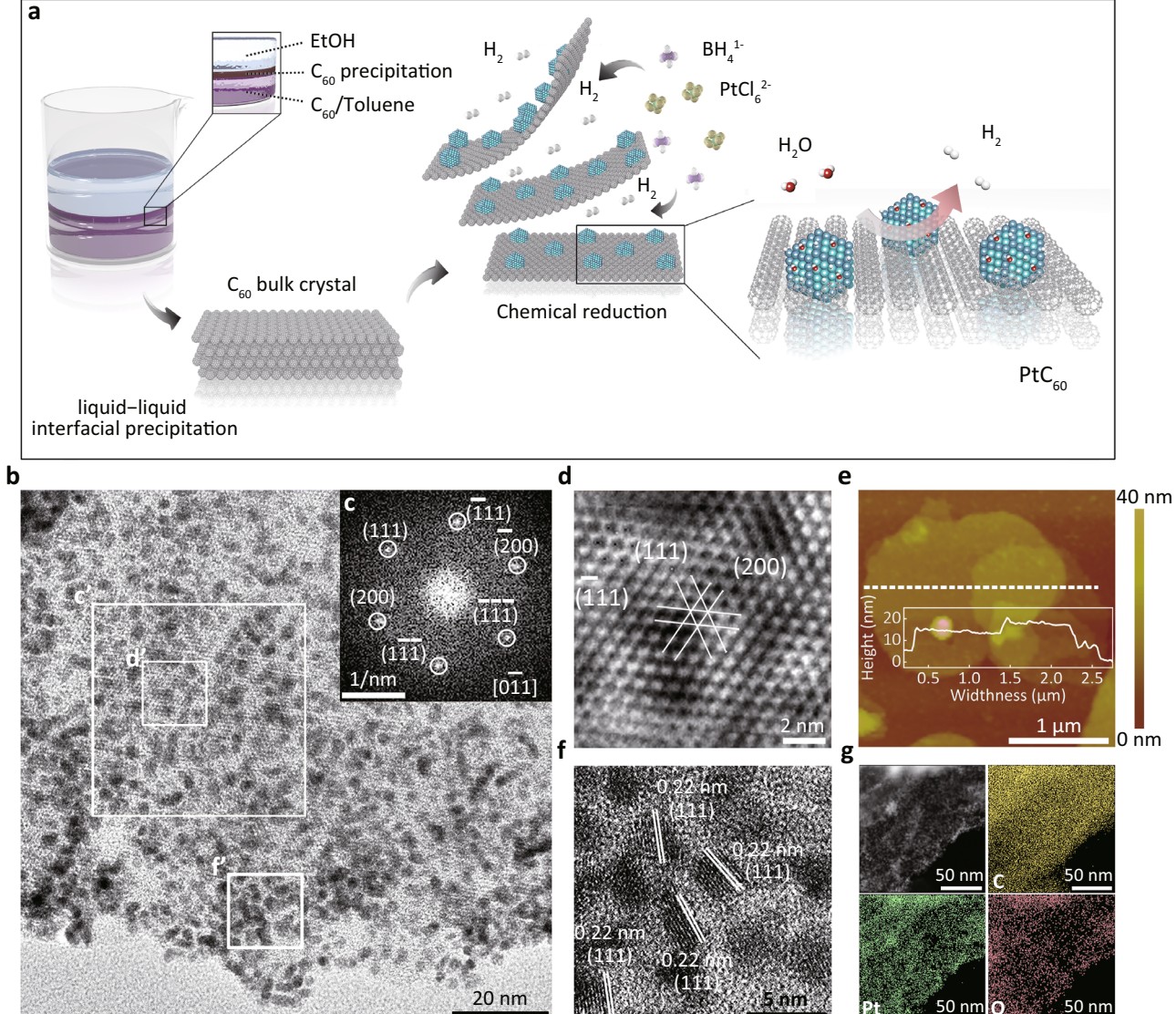

**Fig. 1 | Synthesis and characterization of PtC$_{60}$ catalyst. a** Schematic of the synthetic route of PtC$_{60}$. **b** TEM image of PtC$_{60}$. **c** Corresponding FFT pattern of region c' in (**b**). **d** Corresponding inverse-FFT of region d' in (**b**). **e** AFM image of PtC$_{60}$. **f** High-resolution TEM image of region f' in (**b**). **g** STEM-EDS mapping images of PtC$_{60}$.

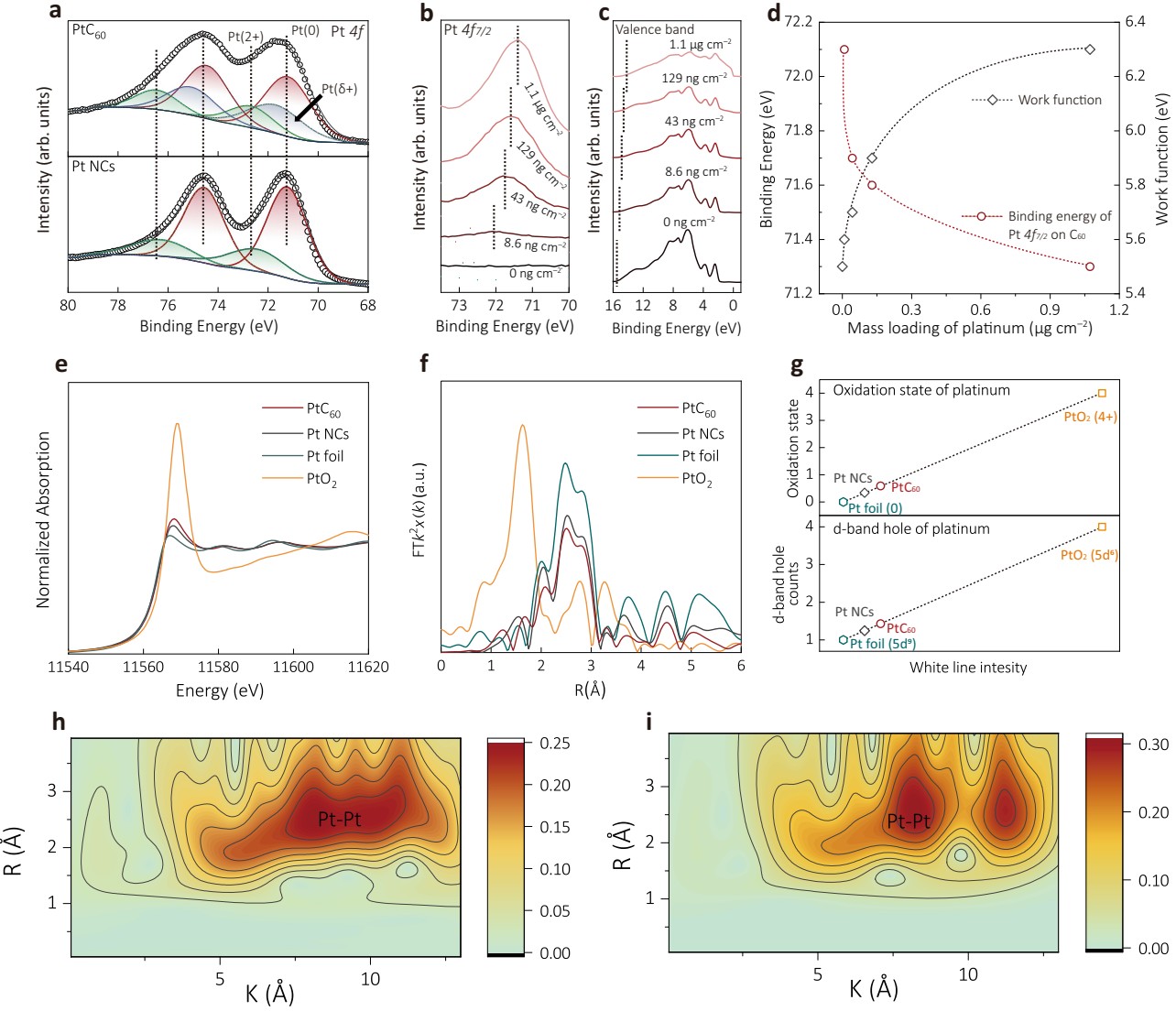

**Fig. 2 | Electronic Characterization and Analysis. a** High-resolution Pt *4f* XPS spectra of PtC$_{60}$ and Pt NCs. **b, c** All-in-vacuum XPS and UPS spectra of Pt/C$_{60}$ film within the same vacuum chamber. **d** Binding energy of Pt *4f$_{7/2}$* and work function depending on the amount of Pt deposited on C$_{60}$ film corresponding to (**b** and **c**).

**e** Normalized XANES spectra at Pt *L$_3$*-edge of PtC$_{60}$, Pt NCs, Pt foil, and PtO$_2$. **f** EXAFS spectra of PtC$_{60}$, Pt NCs, Pt foil and PtO$_2$. **g** Average oxidation states, and d-band hole counts fitted via XANES spectra in (**e**). **h, i** Wavelet transform for the k$^3$-weighted EXAFS spectra for PtC$_{60}$ and Pt NCs, respectively.

sample exposure to oxygen is avoided to the best extent. As shown in Fig. 2b, the initial Pt *4f$_{7/2}$* signal of the Pt (at the loading of 8.6 ng cm$^{-2}$) on C$_{60}$ locates at 72.1 eV, about 1 eV higher than that of the metallic Pt, which is in good agreement with the electron transfer from Pt nanoclusters to C$_{60}$. Along with the deposition of Pt, the binding energies of Pt *4f* electrons decrease to 71.4 eV when the loading of the deposited Pt reaches 1.1 μg cm$^{-2}$ (Fig. 2d). The steep transition of the Pt *4f* binding energy with changing Pt loading suggests that the interactions between Pt and C$_{60}$ are short-ranged, and thus the chemical properties of the Pt atoms that are remote from the Pt/C$_{60}$ interface are not affected by the nanoparticle-support interactions. Moreover, UPS spectra at a low kinetic energy region show the change of the system's work function from 5.5 to 6.3 eV upon Pt deposition on C$_{60}$ (Fig. 2c, d), further confirming the profound charge redistribution at the Pt/C$_{60}$ interface.

X-ray absorption near-edge spectroscopy (XANES) was conducted for the PtC$_{60}$ composite at the Pt *L$_3$*-edge to further study its interfacial electronic structure. As shown in Fig. 2e, the white-line intensity of PtC$_{60}$ is higher than that of Pt nanoclusters and Pt foil, and its oxidation state can be estimated (~0.59) using Pt foil and PtO$_2$ as references

(Fig. 2g). Besides, the d-band hole number for Pt in PtC$_{60}$ was estimated to be 1.44 based on the Pt foil ($5d^96s^1$) and PtO$_2$ ($5d^66s^0$) standards, indicating the presence of unoccupied Pt 5d-orbitals (Fig. 2g). In addition, the extended X-ray adsorption fine structure (EXAFS) (Pt *L$_3$*-edge, Fig. 2f and Supplementary Fig. 19) and the corresponding wavelet transform (Fig. 2h, i and Supplementary Fig. 20) of both PtC$_{60}$ and Pt NCs suggest that no significant Pt-O is present in the samples (Supplementary Table 1). The combined XPS, XAS and UPS data unambiguously confirm the strong character of electronic interactions at the Pt/C$_{60}$ interface. Taken together, the strong confinement effect of C$_{60}$ nanosheets leads to the formation of the Pt clusters with narrow size distribution (2 nm), which is likely originated from the combination of the unique surface morphology of C$_{60}$ nanosheets and the electronic interactions between the Pt atoms and the C$_{60}$ at the interface.

## HER on PtC$_{60}$ in alkaline electrolyte

The alkaline HER on PtC$_{60}$ was studied using a rotating disk electrode in 1.0 M KOH. PtC$_{60}$ with modest loading of ~0.4 mg cm$^{-2}$ reaches the current densities of 10 mA cm$^{-2}$, 50 mA cm$^{-2}$ and

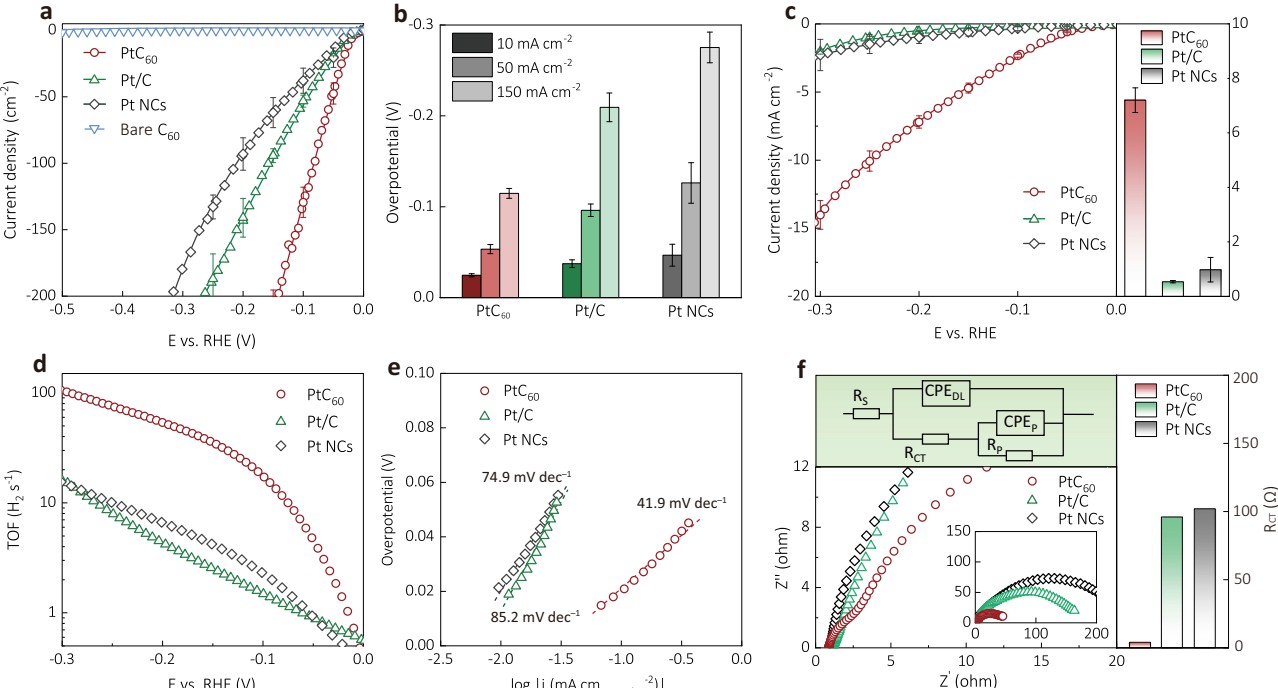

**Fig. 3 | Alkaline HER. a** LSV (linear sweep voltammetry) of Pt catalysts with the loading of 0.4 mg cm$^{-2}$ and bare C$_{60}$. **b** Overpotentials of the catalysts at current densities of 10, 50, 150 mA cm$^{-2}$ in (**a**). **c** LSV of the catalysts with very low catalyst loading of 0.004 mg cm$^{-2}$ for assessing the intrinsic activities. Inset: geometric current density of the samples at an overpotential of 200 mV. **d** TOFs of the catalysts with the catalyst loading of 0.004 mg cm$^{-2}$. **e** Corresponding Tafel plots obtained from LSV in (**c**). **f** EIS Nyquist plots of the catalysts at −40 mV vs. RHE. Inset (top): the equivalent circuit used for the EIS fitting. Inset (right): comparison of the fitted R$_{CT}$. (Sample size $n$ = 3 for (**a**–**d**) error bars correspond to the standard deviation of three independent measurements).

150 mA cm$^{-2}$ at overpotentials of 24.3 mV, 53.2 mV and 110.0 mV, respectively. These overpotentials are much lower than that of commercial 20% Pt/C and Pt NCs under the same conditions, indicating that sites at the Pt/C$_{60}$ interface can promote alkaline HER activity (Figs. 3a, 3b). In addition, PtC$_{60}$ with different Pt loadings were tested for HER under the same conditions, and all displayed enhanced activity compared to those of the Pt/C and Pt NCs (Supplementary Fig. 21). Moreover, negligible degradation was observed for PtC$_{60}$ in chronopotentiometry measurement carried out at 10 mA cm$^{-2}$ for 50 h, while both Pt NCs and Pt/C showed obvious activity loss (Supplementary Fig. 22). The TEM images (Supplementary Fig. 23) of PtC$_{60}$ after catalysis show no noticeable sintering and other morphological changes. We attribute this structural stability to the strong interactions between Pt NCs and the C$_{60}$ nanosheet, as illustrated in the annealing experiment. We also found that the PtC$_{60}$ sample shows higher hydrophilicity (Supplementary Fig. 24) than Pt/C and Pt NCs (Supplementary Note 1), which likely facilitates the bubble evolution and hence prevents the catalyst structure from collapsing during HER[59–61].

Tafel analysis was carried out to investigate the HER kinetics. Similarly, low Tafel slopes (under 30 mV dec$^{-1}$) were obtained for PtC$_{60}$, Pt/C, and Pt NCs at the loading of 0.4 mg cm$^{-2}$ (Supplementary Fig. 25). Such small Tafel slopes would suggest that the Tafel step determines the overall HER rates, which is counterintuitive since the Volmer step is usually recognized as the limiting step for alkaline HER. Thus the above measured HER kinetics is controlled, at least largely, by the mass transportation of H$_2$ generated on the electrode surface[62–64]. The importance of H$_2$ diffusion for HER rate was also corroborated by mass transport modelling (Supplementary Fig. 26, Table 2, and Note 2). Therefore, it is crucial to exclude the mass transport effect and evaluate the intrinsic activities of the catalysts to confirm the rational design of the PtC$_{60}$ composite for efficient alkaline HER.

## Investigation of the intrinsic activities of PtC$_{60}$ towards alkaline HER

To minimize the mass transportation effect, we conducted HER using electrodes with decreased catalyst loadings[62]. As shown in Fig. 3c, with a low catalyst loading of 0.004 mg cm$^{-2}$, PtC$_{60}$ exhibits a much higher current density compared to those of Pt/C and Pt NCs at all potentials. For instance, PtC$_{60}$ affords a geometric current density of 2.3 mA cm$^{-2}$ at the overpotential of 100 mV, which is 11 and 7 times larger than those of Pt/C (0.19 mA cm$^{-2}$) and Pt NCs (0.34 mA cm$^{-2}$). Similar activity trends were observed on electrodes with various catalyst loadings before and after normalization by the Pt mass loading (Supplementary Fig. 27 and 28). Under the same conditions, the mass activity of PtC$_{60}$ is estimated to be 1.55 A mg$^{-1}$, which is much higher than those of the Pt/C (0.24 A mg$^{-1}$) and Pt NCs (0.08 A mg$^{-1}$) at overpotential of 100 mV. Moreover, the HER activity of PtC$_{60}$ also exceeds those of Pt/C and Pt NCs after the normalization by the electrode surface area estimated via under potential deposited H (Supplementary Fig. 29, 30 and Table 3), which was shown to reflect the specific active surface area of Pt-based materials[65,66]. Specifically, the turnover frequency (TOF) of PtC$_{60}$ (17.6 s$^{-1}$) estimated based on the number of surface Pt-sites was remarkably higher than those of Pt/C (1.5 s$^{-1}$) and Pt NCs (2.8 s$^{-1}$) (Fig. 3d) at overpotential of 100 mV. Taken together, we confirm that PtC$_{60}$ shows enhanced intrinsic activity compared to the state-of-the-art-catalyst Pt/C, by more than an order of magnitude in terms of both TOF and mass activity.

Tafel analysis was also conducted at low catalyst loadings to explore the intrinsic kinetics for alkaline HER for the prepared catalysts. Obvious increases in Tafel slopes were observed for both Pt/C (85.2 mV dec$^{-1}$) and Pt NCs (74.9 mV dec$^{-1}$) when decreasing the catalyst-loading (Fig. 3e, Supplementary Fig. 31b, c), suggesting that the H$_2$ mass transport became less dominant, and the HER rate is determined by a mix of Volmer step and Heyrovsky step. In contrast, the Tafel slope of PtC$_{60}$ (41.9 mV dec$^{-1}$) increased insignificantly at low

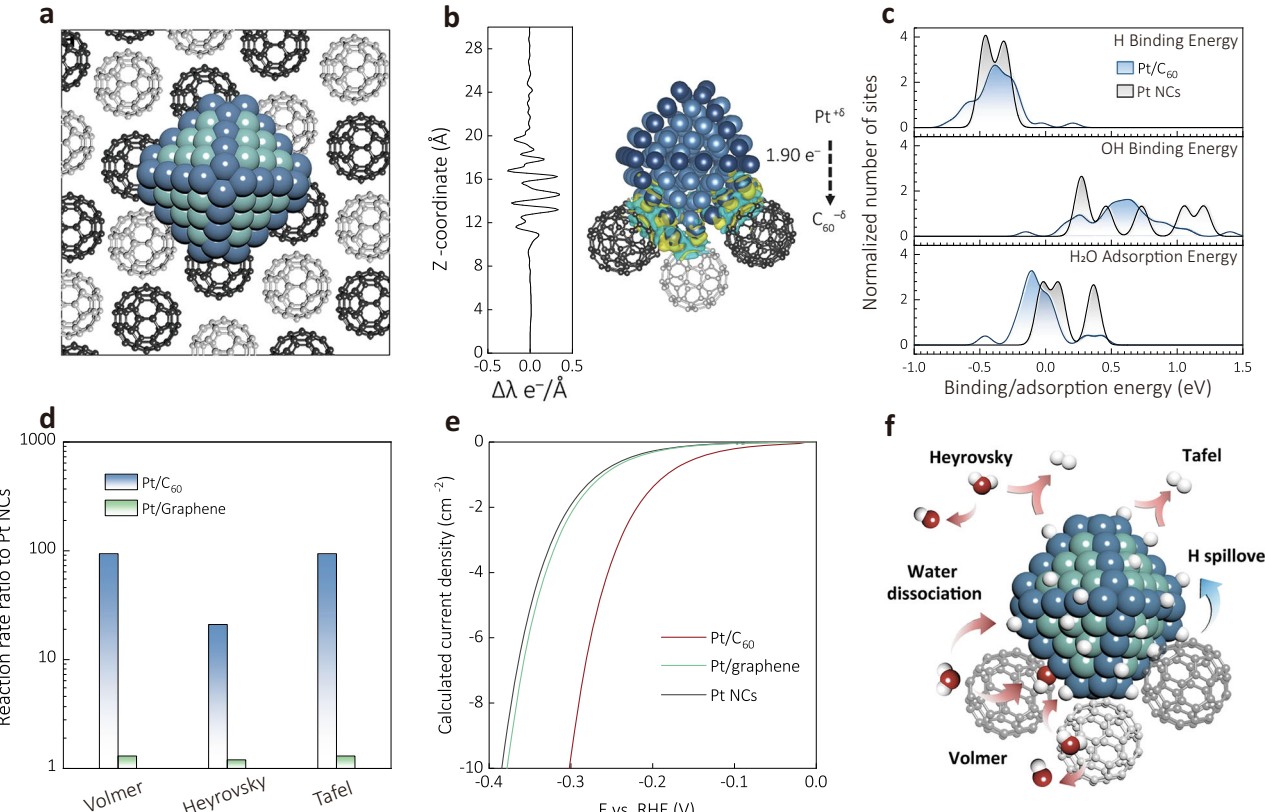

**Fig. 4 | Atomistic insight. a** The employed $Pt/C_{60}$ model with the C atoms in the top and bottom fullerene layers displayed in black and grey, respectively, Pt atoms on edge in dark blue and Pt atoms on terraces in lighter blue. **b** Electronic density difference due to the interactions between Pt and $C_{60}$ as an integrated 1D profile and 3D-isosurface plotted at 0.02 e $Å^{-3}$ value. Yellow and cyan regions of the iso-surface represent excess and deficit of electrons, respectively. **c** Distribution of binding Gibbs free energies of H (top), OH (middle), and $H_2O$ (bottom) on

unsupported Pt NCs compared to the much wider distributions calculated on $Pt/C_{60}$. **d** The enhancement in the reaction rate constants of the various alkaline HER reaction steps on $Pt/C_{60}$ (011) and Pt/graphene relative to the unsupported Pt NCs estimated through activation energies. **e** LSV of supported $Pt/C_{60}$, Pt/graphene, and unsupported Pt NCs calculated from the microkinetic model. **f** Summary of the reaction steps on the $Pt/C_{60}$.

catalyst loading (Supplementary Fig. 31a), suggesting that the HER is mostly limited by the Heyrovsky step, similar to the HER kinetics in acidic conditions where the Volmer step is fast enough (Supplementary Fig. 32)[59,60]. Moreover, electrochemical impedance spectroscopy (EIS) revealed a much lower charge transfer resistance $R_{CT} = 4.0$ ohm for $PtC_{60}$ electrode corresponding to H adsorption compared to those of the Pt/C (96.0 ohm) and Pt NCs (102.0 ohm) electrodes, confirming the accelerated water dissociation or Volmer step on the $PtC_{60}$. (Fig. 3f, Supplementary Fig. 33, and Table 4).

**Theoretical exploration of the enhanced HER activity on $PtC_{60}$**
The mechanistic insights into the effect of Pt-$C_{60}$ interactions on the HER activity were further corroborated by density functional theory (DFT) calculations and microkinetic analysis using the model of Pt nanocrystallites (1.5 nm size) supported on (011) surface of crystalline $C_{60}$ (Fig. 4a). Such nanocrystallites have fcc crystal lattice similar to the clusters observed by TEM (Fig. 1b-f) and were previously demonstrated to reliably represent the properties of experimentally prepared nanoclusters[67,68]. First, we screened thousands of tentative structures for the $Pt/C_{60}$ interface using Modified Embedded-Atom Method (MEAM) interatomic potentials[69], to obtain the realistic $Pt/C_{60}(011)$ models. Then, 16 different structures with the lowest obtained energies were further optimized *via* PBE + D3 density functional to identify the most energetically stable interface configuration (Fig. 4a, Supplementary Fig. 34, and Note 3–4)[70,71]. Bader analysis shows that the metal-support electronic interactions lead to the transfer of 1.90 electrons from the Pt cluster to the $C_{60}$ plane, resulting in ~ +0.1 charge on Pt

atoms at the $Pt/C_{60}$ interface (Supplementary Fig. 35). Although Pt donates electrons to $C_{60}$ support, a detailed analysis of the electronic density polarization in $Pt/C_{60}$ reveals 7.1 Debye dipole moment pointing towards the fullerene support per Pt cluster (Fig. 4b, Supplementary Note 5)[72]. In line with the experimental observations (Fig. 2d), such dipole moment is calculated to significantly increase the work function of $Pt/C_{60}$ to 6.00 eV compared to 5.25 eV for unsupported Pt (Supplementary Fig. 36 and Note 6). As shown in Supplementary Fig. 37. the C atoms on $C_{60}$ contacting with Pt receive a small but noticeable amount of electrons ($\Delta Q = 1.18$ $e^-$). Consequently, the electronic densities of states (DOS) of these C-atoms exhibit a more disperse distribution (Supplementary Fig. 37a) compared to that of the C-atoms that are further away from the Pt (Supplementary Fig. 37b). Note, such dramatic differences in DOS were not observed in the Pt/graphene system (Supplementary Note 6, Supplementary Fig. 37 c, d), whose structure was similar to the Pd/graphene model designed previously. Thus, the DOS analysis further confirms the profound electronic interactions at the $Pt/C_{60}$ interface, in contrast to the weak interfacial interactions at the Pt/graphene interface as reported in the previous literature[73–75].

We further assessed the improvement of HER activity of $Pt/C_{60}$ compared to the unsupported Pt NCs and Pt NCs supported on graphene through an electrochemical activity analysis based on a recently developed microkinetic model for alkaline HER on Pt(111)[76]. First, the differences between the highest and the lowest binding energies of H and $H_2O$ species were calculated to be 0.45 eV and 0.89 eV for Pt sites at the $Pt/C_{60}$ interface as compared to 0.17 eV and 0.39 eV for Pt-sites

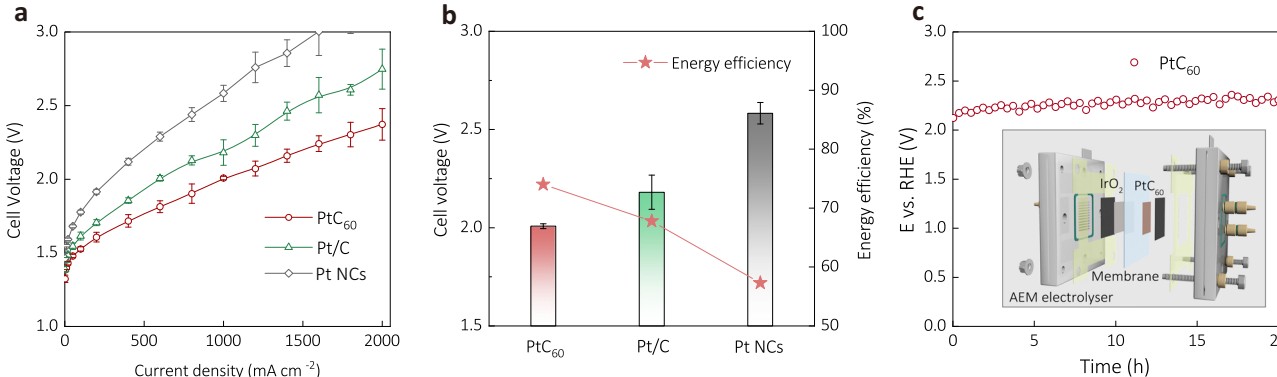

**Fig. 5 | Performance of AEM electrolysers. a** LSV curves of the AEM electrolysers using $PtC_{60}$||$IrO_2$, PtC||$IrO_2$ and Pt NCs||$IrO_2$, respectively. **b** Cell voltage at the current density of $1\,A\,cm^{-2}$ and the corresponding energy efficiency. **c** Durability test of the AEM electrolyser at the current density of $1\,A\,cm^{-2}$. (Sample size $n = 3$; error bars correspond to the standard deviation of three independent measurements).

on unsupported Pt NCs, and 0.19 eV and 0.42 eV for sites on Pt/graphene interface, respectively (Fig. 4c, Supplementary Note 7, Table 5 and 6). This profound difference suggests much more heterogeneous properties (in terms of binding strength towards key reaction intermediates) for Pt sites at the $Pt/C_{60}$ interface compared to the sites on Pt/graphene. As a result, some of the $Pt/C_{60}$ interface sites exhibit low activation energies for Volmer step, whereas other sites on the interface have low activation energies for Heyrovsky and/or Tafel steps (Supplementary Fig. 38, 39, Table 7, 8 and Note 8). After averaging all Pt-sites at the $Pt/C_{60}$ interface, the rates of the Volmer, Heyrovsky, and Tafel steps on $Pt/C_{60}$ are calculated to be 94, 21, and 94 times higher than the respective rates on the unsupported Pt NCs, or 90, 15, and 90 times higher than those on Pt/graphene (Fig. 4d). Note that H atoms are calculated to be able to freely move diffusion barriers as low as 0.22 eV between sites that are highly active in various reaction steps, which could increase HER activity even further (Fig. 4f, Supplementary Fig. 40 and Note 8). In line with the experimental results (Fig. 3), our analysis based on the microkinetic model shows that the $Pt/C_{60}$ composite requires much smaller overpotential to achieve a current density of $10\,mA\,cm^{-2}$ compared to the unsupported Pt and the Pt/graphene, by 0.09 V and 0.08 V respectively (Fig. 4e). The enhancement becomes much more dramatic if we evaluate normalized activity only for Pt-sites at the $Pt/C_{60}$ interface (Supplementary Fig. 41). Overall, our model reveals the origin of the improved activity of $Pt/C_{60}$ towards alkaline HER activity, that is the diversity of binding properties of the Pt-sites at the $Pt/C_{60}$ interface.

## Alkaline HER on $PtC_{60}$ in an industrially relevant electrolyser
To test $PtC_{60}$ under more practical conditions, we constructed a membrane-electrode-assembly (MEA) based electrolyser based on a recent protocol[77], using AEM as the solid electrolyte and commercial $IrO_2$/C as the anodic catalyst (Fig. 5c and Supplementary Fig. 42). Under the same testing conditions, the same activity trend of $PtC_{60} >$ Pt/C > Pt NCs (Fig. 5a) was observed again in the electrolysers. Specifically, the electrolyser using $PtC_{60}$ reaches a current density of $1.0\,A\,cm^{-2}$ at a cell voltage of 2.01 V, which is much lower than those of the electrolysers with Pt/C (2.18 V) and Pt NCs (2.58 V) catalysts (Figs. 5a, 5b). Consequently, the energy efficiency of the $PtC_{60}$-based AEM electrolyser reaches 74% at $1\,A\,cm^{-2}$ (Supplementary Table 11, Note 9), surpassing those of Pt/C (67.8%) and Pt NCs (57.3%) containing electrolysers (Fig. 5b)[78]. Moreover, we operated the $PtC_{60}$-based AEM electrolyser at $1\,A\,cm^{-2}$ for a 20 h-long stability test and observed minimal activity loss (Fig. 5c). We believe that the performance of the above electrolysers can be further improved with optimizations on other components, *e.g.*, substrate, membrane, cell geometries, temperature, etc., which is out of the scope of this work.

## Discussion
In this work, two-dimensional $C_{60}$ nanosheets were developed for anchoring Pt-nanoclusters and producing heterostructures with remarkably high activity toward alkaline HER. Comprehensive physical characterizations, density functional simulations, and kinetic analysis revealed that electron polarization at $Pt/C_{60}$ interface introduces significant variation in binding properties of the active sites, which on average become more active in all steps of HER than sites on unsupported or graphene-supported Pt NCs. Meanwhile, the diffusion barriers of adsorbed hydrogen are calculated to be as low as 0.22 eV on Pt enabling hydrogen spillover among sites with high activity for different elementary steps, ensuring the enhanced HER activity. In addition, the strong confinement of Pt-nanoclusters on $C_{60}$ nanosheets prevents the sintering of the Pt-nanoclusters during catalysis and further affords good stability. Finally, $PtC_{60}$ delivers promising performance in an AEM electrolyser operated under practically relevant testing conditions. Overall, we believe that our design strategy of introducing hetero-interface with diverse binding energies of the key reaction intermediates can be applied broadly to other energy applications that depend on the high performance of electrocatalysis.

## Methods
### Materials and chemicals
Ethylene glycol (EG, anhydrous, 99.8%), toluene (anhydrous, 99.8), Ethanol (reagent grade, 99%+), chloroplatinic acid ($H_2PtCl_6 \cdot xH_2O$, 99.9%), sodium borohydride ($NaBH_4$, purum p.a., 96%+), $IrCl_3 \cdot 3H_2O$ (reagent grade, 99.9%), potassium hydroxide (flakes, 90%), Nafion®117 containing solution (5% in a mixture of lower aliphatic alcohols and water), 2-Propanol (anhydrous, 99.5%) and Pt/C catalyst (20 wt.% loading on Vulcan XC-72) were purchased from Sigma-Aldrich. Co. Commercial Ir/C catalyst (20 wt.% loading on Vulcan XC-72) was purchased from Premetek. Co. Buckminsterfullerene ($C_{60}$, 99%) powder was purchased from TanFeng. Int. XC-72 carbon black was purchased from Suzhou Sinero Technology Co., Ltd. All chemicals were used in their as-received condition without further purification.

### Preparation of $C_{60}$ bulk crystal
The preparation of $C_{60}$ bulk crystal was based on the liquid-liquid-interface-precipitation method. Typically, 20 mg $C_{60}$ powder was dissolved in 2 mL toluene and ultrasonicated for 10 min. Next, 2 mL Ethanol was added slowly to the mixed solution to form a liquid-liquid interface, and then the solution was transferred to a refrigerator and kept at 5 °C for 24 h. Caution is needed to maintain the liquid-liquid interface. The sample was collected by centrifuge and dried in an oven at 80 °C for 10 h.

## Preparation of Pt nanoclusters on $C_{60}$ nanosheets (PtC$_{60}$), pristine Pt nanoclusters (Pt NCs), and Pt nanoclusters on XC-72 carbon black (Pt/CB)

The PtC$_{60}$ was prepared via a facile solution-phase method. In a typical preparation, 5 mg of $C_{60}$ bulk crystal was added into 2 mL ethanol and ultrasonicated for 30 min, then 50 μL of $H_2PtCl_6$ . $x$H$_2$O/EG solution (1.0 M) and 10 mL EG were added into the mixture, and the mixture was kept at stirring for 12 h. Subsequently, 200 mg NaBH$_4$ was added into the above mixture slowly, and the obtained mixture remained stirred for 12 h. Then the samples were collected via centrifugation and dried in an oven at 80 °C for 12 h. Inductively coupled plasma mass spectrometry (ICP-MS, PerkinElmer Optima 5300DV) was carried out to determine the actual mass ratio of Pt and $C_{60}$ that was 37.5 wt%. Besides, the loading of Pt on the $C_{60}$ was controlled by adjusting the amount of $H_2PtCl_6$ (e.g., 35 μL and 100 μL of the $H_2PtCl_6$/EG (1.0 M) solution), and the contents of Pt were 32.5 wt% (PtC$_{60}$–32.5 wt%) and 47.5 wt% (PtC$_{60}$–47.5 wt%), respectively. As a comparison, bare Pt nanoclusters (Pt NCs) were synthesized under the same condition in the absence of $C_{60}$ bulk crystal, and Pt nanoclusters on XC-72 carbon black (Pt/CB) were prepared using the same method for preparing PtC$_{60}$ (37.5%) while $C_{60}$ was substituted by carbon black.

## Materials characterization

Transmission electron microscopy (TEM) images were captured by a JEOL JEM-2010 TEM (working voltage: 200 kV), and high angle annular dark-field scanning transmission electron microscopy (HAADF-STEM) images were obtained on a JEM-ARM 200 F at an operating voltage of 200 kV. The atomic force microscopy (AFM) was performed by a Dimension FastScan, Bruker Corp., USA, operating in a tapping mode. The X-ray diffraction (XRD) analysis was carried out by a GBC MMA X-ray diffractometer with a Cu Kα irradiation source (λ = 1.54056 Å). The X-ray photoelectron spectroscopy (XPS) analysis was done using a Thermo ESCALAB 250 (monochrome Al Kα, hv = 1486.6 eV). The Fourier-transform infrared spectroscopy (FT-IR) analysis was carried by a Shimadzu FT-IR Prestige-21 spectrometer (KBr as the background). The contact angle analysis was conducted with Data physics OCA15 and used 3 μL 1 M KOH as the solution.

The all-in-vacuum UPS and XPS measurements (including the Pt atomic layer deposition and $C_{60}$ thin film deposition) were all performed in our customized ultrahigh vacuum chamber (~10$^{-10}$ mbar), using He I (21.2 eV) and Mg Kα (1253.6 eV) as excitation sources, respectively. Using a sample bias voltage of −7 V, the work function of the sample was obtained by the secondary electron cut-off in the low kinetic energy region using the following Eq. (2)[79,80]. (Supplementary Fig. 43a) The Fermi level was calibrated with respect to the sputter-cleaned gold foil measured at room temperature. The mass loading of the Pt was estimated by measuring the attenuation of C 1 s peak before and after Pt deposition and then calibrated by a quartz crystal microbalance (QCM) located in front of the sample stage.

$$\phi = hv - W \tag{2}$$

$$\phi = hv - (E_{SECO} - E_F) \tag{3}$$

W is the spectrum from $E_F$ to $E_{SECO}$. $hv$ is 21.21 eV (He). $E_F$ is 0 eV, $E_{SECO}$ is 15.7 eV, 15.6 eV, 15.5 eV, 15.3 eV, and 14.9 eV for 0 nm, 0.004 nm, 0.06 nm, 0.02 nm, and 0.5 nm Pt average thickness (Supplementary Fig. 43b). Hence, the work function calculated via Eq. (3) is 5.5 eV, 5.6 eV, 5.7 eV, 5.9 eV, and 6.3 eV, respectively.

The X-ray absorption fine structure spectra (XAFS) Pt L-edge measurements were performed at BL07A1 beamline of National Synchrotron Radiation Research Centre (NSRRC). The data of PtC$_{60}$ were collected in fluorescence mode using a Lytle detector, while the Pt NCs were collected in transmission mode. The sample was ground and uniformly daubed on the special adhesive tape. In addition, Pt foil and commercial PtO$_2$ powder were used as references for Pt$^0$ and Pt$^{4+}$ valence states, respectively. The XANES and EXAFS data were analysed using Athena and Artemis software, respectively. EXAFS data was used to extract the coordination number (N), bond distance (R, in units of Å), and Debye-Waller factor (2).

## Electrochemical measurements in a three-electrode system

Electrochemical measurements were performed using a computer-controlled potentiostat (Biologic, VSP-300) in a typical three-electrode cell with a rotating disk electrode (RDE, Pine Research Instruments) at room temperature. A Hg/HgO (1 M) and a graphite rod were used as the reference electrode and the counter electrode, respectively. For the preparation of catalyst ink, 2.5 mg of the catalyst was dispersed into a mixture of 384 μL of deionized water, 100 μL of 2-Propanol and 16 μL of 5 wt% Nafion solution, and the obtained ink was ultrasonicated for 3 h before use. In a typical process of preparing the working electrode, 10 μL of the catalyst ink was loaded onto a glassy carbon electrode (0.1256 cm$^2$) and dried for 20 min at room temperature (0.4 mg cm$^{-2}$ of catalyst). For the low-catalyst-loading measurement, the as prepared catalyst ink was diluted 10 and 100 folds by Ethanol, and then 10 μL of the diluted inks were load onto a glassy carbon electrode for the loading of 0.04 mg cm$^{-2}$, and 0.004 mg cm$^{-2}$, respectively. The rotating speed of the working electrode is set at 1600 rpm for all experiments. Prior to HER activity investigation, 30 cycles of cyclic voltammetry (CV) from −0.025 V to 0.075 V vs. RHE at a scan rate of 50 mV s$^{-1}$ were run in an H$_2$-saturated 1 M KOH and/or 0.5 M H$_2$SO$_4$ solution for electrochemical cleaning. LSV curves were recorded at a scan rate of 5 mV s$^{-1}$ to evaluate the HER activity. 95% $i$R-compensation was applied to correct the ohmic potential drop. The electrochemical impedance spectra (EIS) were collected at −0.05 V vs. RHE in the frequency range of 0.1 to 100 kHz.

The electrochemical surface area (ECSA) of Pt was estimated by measuring the hydrogen underpotential deposition (HUPD) features. The as-prepared catalysts were tested in the three-electrode setup in an N$_2$-saturated 0.1 M KOH solution, and CV curves from 0 to 0.7 V vs. RHE were collected at a scan rate of 50 mV s$^{-1}$ with a rotating speed of 1600 rpm. The ECSA is calculated using the following Eq. (3) from the region of hydrogen adsorption charge (-0–0.5 V) on CVs (negative-going potential scan, around 0 to 0.075 V) with the correction for double-layer charging:

$$ECSA_{Pt} = \frac{S_{Q-adsorption}(A \cdot V)/v(Vs^{-1})}{210(\mu c\ cm_{Pt}^{-2})} \tag{3}$$

$S_{Q\text{-adsorption}}$ is the integral area of the region of hydrogen adsorption charge on the CV. $v$ is the scan rate, and 210 μC cm$^{-2}$ is used as the conversion factor[81,82].

The TOF values were calculated based on the number of surface Pt atoms in each sample electrode according to the following equation:

$$TOF = \frac{1}{2} \cdot \frac{I(A)/F(C\ mol^{-1})}{S_{Q-adsorption}(A \cdot V)/v_{scanrate}(Vs^{-1})F(C\ mol^{-1})} \tag{4}$$

where I (in A) is the current recorded from the LSV curves, and F is the Faraday constant (in C mol$^{-1}$).

## Computational methods

The Large-scale Atomic/Molecular Massively Parallel Simulator (LAMMPS) package[83] and Modified Embedded Atom Method (MEAM) potential[69] were utilized to screen tentative structures of Pt/C$_{60}$(011) interfaces. The C$_{60}$(011) surface was modeled by a slab with two C$_{60}$ layers and a thickness of 13 Å. The employed $\sqrt{8} \times \sqrt{8}$ supercell had 24.5 × 24.5 Å lateral dimensions, with a vacuum of 12 Å, allowing for

separation between supported Pt NCs to exceed 9 Å, which is essential for adsorption energies to be unaffected by particle-particle interactions. The Pt NC was translated and rotated about the different two axes via a systematic *11×11×11* grid scan to identify the most realistic and energetically stable structure. The calculations were performed keeping the surface fixed to keep the lattice parameters and avoid the distortion of the slab. The threshold parameters for geometry optimization in LAMMPS calculations were $1×10^{-8} eV$ for the energy and $1×10^{-8} eV/A$ for the forces.

DFT calculations of the most stable Pt/$C_{60}$(011) structures obtained with MEAM were performed using the Vienna Ab-initio Simulation Package (VASP)[84-86] and Perdew Burke Erzenhof (PBE)[87] functional. In turn, the dispersive van der Waals interactions are also considered by means of the D3 approximation developed by Grimme et al.[70,71]. The Methfessel-Paxton smearing method with the smearing width of 0.1 eV was employed to set the partial occupancies for each orbital. The interactions between core and valence electrons were described with the projector augmented wave (PAW) approach. A plane-wave basis set with the cut-off energy of 400 eV was used. The calculations involving the Pt NCs were performed at the gamma point. The threshold for the SCF calculations was set to $1×10^{-5} eV$ for the changes in the energy, while the geometric relaxations were terminated once the forces acting on all atoms were smaller than $3×10^{-2} eV/A$. More information on the LAMMPS and DFT calculations can be found in Supplementary Information Note 4–5 and Supplementary Figs. 34, 35.

The adsorption DFT energies of the different species on substrate are calculated through the following generic Eq. (5) shown below.

$$E_{ads}(adsorbate - substrate) = E(absorbate - substrate) \\ - E(absorbate) - E(substrate) \quad (5)$$

The Gibbs free energies of the considered species are calculated in Atomic Simulation Environment (ASE) using the ideal gas thermodynamic model and the harmonic thermodynamic model[88]. More information on the DFT energies and Gibbs Free energies calculations can be found in Supplementary Information Note 5 and 7, respectively. The microkinetic analysis was performed employing the binding Gibbs Free energies of $H_2O$, H, and OH on different Pt active sites. The barriers for different reaction steps were extrapolated from the previous analysis of alkaline HER on Pt(111) in Ref. [70,89]. The detailed analysis together with the data for the different reactions and activation energies for the different Pt sites are collated in Supplementary Note 8.

**Electrochemical measurements in AEM electrolyser**
The AEM testing is based on a recent protocol[77]. First, the as-received ion-exchange-resin membrane (SELEMION AMN/N Type1) was immersed into 1.0 M KOH for 24 h before the construction of the AEM electrolyser. Pt$C_{60}$, commercial Pt/C and Pt NCs were used as the cathode catalysts. The catalyst ink for Pt$C_{60}$ was prepared in the same way as described above. 1.5 mg of Pt$C_{60}$ was deposited onto a nickel foam with surface area $1×1 cm^2$. The AEM electrolyser was evaluated at 60 °C, using 1.0 M KOH as the electrolyte with a flowing rate of 40 ml/min. The IrO$_2$ on titanium foam was prepared as the anode for the electrolyser. Briefly, 400 ml of IrCl$_3$·3H$_2$O ethanol solution (10 mg/ml) was deposited on titanium foam with surface area of $1×1 cm^2$. Then the titanium foam loaded with IrCl$_3$ was calcinated in a muffle furnace at 400 °C for 30 min. Prior to AEM testing, 10 cycles of cyclic voltammetry (CV) were conducted from 0 V to 1.5 V of cell voltage at a scan rate of 50 mV s$^{-1}$. Then the AEM electrolyser was conducted at 2 mA cm$^{-2}$ for 5 min to stabilize. Subsequently, the electrolyser was tested at 2 mA cm$^{-2}$, 10 mA cm$^{-2}$, 20 mA cm$^{-2}$, 50 mA cm$^{-2}$, 100 mA cm$^{-2}$, 200 mA cm$^{-2}$, and increased in 200 mA cm$^{-2}$ steps until reaching 2 A cm$^{-2}$ via a chronopotentiometry method, and the potential was recorded,

measuring the potential for 15 s at each step to collect the current-voltage curve. For the stability measurement, the catalyst ink was prepared in the same way as above. Then, 3.0 mg of Pt$C_{60}$ and 3 mg of Ir/C were deposited onto the nickel foams employed as cathode and anode with an active surface area of $1×1 cm^2$, respectively. The CP test was carried out at 60 °C, using 1.0 M KOH as the electrolyte with a flowing rate of 40 mL/min.

## Data availability
The data that support the findings of this study are available from the Supplementary Information and/or from the corresponding author upon reasonable request.

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

## Acknowledgements

We acknowledge the National University of Singapore, Ministry of Education for their financial support, through the grants of A-0009176-02-00 and A-0009176-03-00, A*STAR (Agency for Science, Technology and Research) under its LCERFI program (Award No U2102d2002), Centre for Hydrogen Innovations at NUS (CHI-P2022-06). L. Wang and S. M. Kozlov would also like to acknowledge the support by National Research Foundation (NRF) Singapore, under NRF Fellowships (NRF-NRFF13-2021-0007 and NRFF13-2021-0126). W. Sun acknowledge the support by the Natural Science Foundation of Zhejiang Province (Grant No. LZ22B030006). Computational work was performed using resources of the National Supercomputing Centre, Singapore.

## Author contributions

L.W. conceived and supervised the project. J.C., L.W., and W.S. designed the experiments. J.C. carried out the catalyst performance evaluation and the physical characterizations. M.A., F.B.Z. performed the DFT computations and relative data analysis under the supervision of S.M.K. X.L. performed the XPS and UPS measurements during Pt deposition on C60 film under the supervision of W.C. T.Z. and L.Wen helped conduct the MEA performance testing. Y.Z. performed mass transport modeling. H.Z. helped perform FT-IR measurement. J.C., M.A., F.B.Z., S.M.K., and L.W. analysed the data and prepared the manuscript. All the authors discussed the results and assisted during the manuscript preparation.

## Competing interests

The authors declare no competing interests.
