## [Peer Review File · Nature Communications]

REVIEWER COMMENTS

Reviewer #1 (Remarks to the Author):

The authors reported a PGM-carbon heterostructure electrocatalyst that the Pt clusters are loaded on the 2D fullerene nanosheets for alkaline hydrogen evolution reaction. One good point mentioned in this work is that they carefully consider the diverse activity of different Pt sites among the whole cluster based on the DFT calculation. However, several unclear and contradictory descriptions in the manuscript still require to be explained. For the AEMWE performance, they need to be extremely improved. I think it can only be considered to be published in Nat. Commun. after major revision. Some comments are given below:

(1) Many Pt/carbon-based heterostructure electrocatalysts have been reported in the literature. Why to choose the fullerene (C60) as the support materials in this work? If considering the interfacial diversity, the strategies such as nanostructuring and heteroatom-doping would increase the sites diversity of most of carbon materials in general. Besides, the diversity of carbon sites is not discussed in the calculation parts. Please give more discussions about the innovation of this work.

(2) In Fig.2 b and c, the authors used the term 'in-situ' to introduce the XPS and UPS spectra. Since the deposition and X-ray measurement is not processed at the same time, thus it is inappropriate (regardless of whether to keep the sample in the high vacuum or not). Furthermore, using thickness to distinguish different samples is not a right way; the 'Pt loading' is better. In fact, the Pt has an atomic radius of 0.139 nm. Samples with thickness from 0.004 to 0.06 nm are not even achieve one single layer of Pt atom, which means they only reflect the coverage of Pt atom on C60 film. Therefore, the negative shift of binding energy with the increase of Pt loading is due to the higher loading of metal Pt⁰ sharing the electron transfer from the C60 film. More detailed, for the 0.004 nm sample, its binding energy is ~72 eV, which is still lower than that of Pt²⁺ (72.5 V). Therefore, it could be more persuasive to speculate that the high valence is formed during the synthesis process, where some of the Pt^{x+} can be stabilized by the C60 due to the support effect. Please give some explanations!

(3) The fitting procedure for UPS spectra should be introduced in the manuscript.

(4) Considering that it is not easy to control the extreme low loading of catalysts, the author should provide the error bars for all the electrochemical performance in Fig. 4b, c.

(5) Anion exchange membrane is the solid 'polymer' electrolyte. For AEM water electrolysis, the chronopotentiometry (CP) method is more common and has more advantages in precisely evaluating performance than the linear sweep voltammetry (LSV) method. The cell performance measured by LSV with a fast scan rate would be limited by the relative slow mass transfer process, where the fast voltage change doesn't allow the reacting system being stable, thus showing an inaccurate and less reproducible current-voltage real-time response. By applying multiple CPs to

record the stable current-voltage response, the polarization curve of the cell can be achieved (Adv. Mater. 2022, 34, 2203033). Moreover, the iR compensation is unnecessary for the AEMWE testing because the resistance would change with the applied current density. As a result, the fixed resistance to plot the linear ohmic loss in Fig. 5d-f might be imprecise.

(7) At 60°C, the thermodynamic potential of water splitting is not 1.23 V anymore. Please check the overpotential listed in the Fig. 5b. Again, error bar should be provided.

Reviewer #2 (Remarks to the Author):

This manuscript reports a new catalyst composed of Pt nanoclusters anchored on C60 sheets for enhanced HER performance. The methods are sound and the analyses and results are representing the cutting edge research. In particular, the mechanistic investigation helps to understand the obtained performance and the industrial devices demonstrate the commercialisation potential. I would recommend publications after addressing the following questions:

1. The main challenge in alkaline membrane water electrolysis is the durability and ion conductivity of the membrane. The authors need to justify the relevance of catalyst development in this broad research context.
2. While the activity of HER catalyst is a problem in alkaline electrolysis, the use of transition metals rather than Pt is often applied due to the relatively good stability and low cost of transition metals in alkaline media. The authors thus need to justify why Pt is of interest in this study.
3. P3, what is the difference between the Volmer step and the water dissociation step in alkaline HER?
4. P3, more details are needed on the C60 crystals, e.g., what is the interaction that holds the C60 together into a flat 2D sheet rather than collapsing into agglomerates? In addition, as the exfoliation is caused by vigorous chemical reaction, how uniform could the sheets be?
5. P4, what is the origin of the strong confinement effect in regulating the narrow size distribution (2 nm) of Pt clusters?
6. P6, I would suggest to move the WY contour in Fig. S17 to the main text. They provide clear information on the bonding states among different samples.
7. P8, more details are needed to explain the mass transfer effect as shown in Fig. S25.
8. P10, is it the binding energy of H or OH in Fig. 4c?

Reviewer #3 (Remarks to the Author):

A contemporary challenge in the electrocatalytic hydrogen production is the development of catalysts for accelerating alkaline HER by improving the intrinsic activity of Pt. This is where the current work contributes by constructing Pt-sites at Pt/C60 interface with diverse binding properties at key-reaction intermediates of HER and achieving enhanced activity. Thus, the work is novel and certainly can be considered for publication in a prime journal. However, the following issues require the attention of the authors to clarify vague points of the manuscript and enhance its quality:

(1) What is the thickness of the bulk C60 crystals, obtained via the liquid-liquid interfacial precipitation method ?

(2) It is not clear how C60 ended up carrying oxygenated species. What is the purity of commercially available C60 used and what are its spectroscopic characteristics based on IR and XPS ? This information is important and it is needed for comparison purposes with the C60 bulk crystals and PtC60, showing the presence of carboxylic acid and hydroxyl moieties.

Overall, major revisions are necessary and only if the authors sufficiently address the aforementioned points, the work can be recommended for publication.

Response to Reviewer 1:

The authors reported a PGM-carbon heterostructure electrocatalyst that the Pt clusters are loaded on the 2D fullerene nanosheets for alkaline hydrogen evolution reaction. One good point mentioned in this work is that they carefully consider the diverse activity of different Pt sites among the whole cluster based on the DFT calculation. However, several unclear and contradictory descriptions in the manuscript still require to be explained. For the AEMWE performance, they need to be extremely improved. I think it can only be considered to be published in Nat. Commun. after major revision. Some comments are given below:

We really appreciate the in-depth comments and constructive suggestions from the referee, particularly the guidance on the AEMWE testing. We have carefully addressed your comments, suggestions and concerns, and made corresponding revisions to the manuscript, which are also described below in a point-by-point fashion. All changes have been highlighted in the Revised Manuscript and Supplementary Information files as well.

1. Many Pt/carbon-based heterostructure electrocatalysts have been reported in the literature. Why to choose the fullerene (C₆₀) as the support materials in this work? If considering the interfacial diversity, the strategies such as nanostructuring and heteroatom-doping would increase the sites diversity of most of carbon materials in general.

Response: We thank the referee for this great point. We fully agree with the referee, nanostructuring and heteroatom-doping are both effective strategies in creating interfacial diversity¹⁻³, and definitely worth trying in the future. However, it may not straightforward to introduce carbon-nanostructures with the extremely small sizes (*i.e.*, ~2 nm), and/or introduce high densities of heteroatom-doping onto a given carbon substrates, which may lead to that only limited Pt nanoparticles can experience these effects. In contrast, the original goal of this study was to achieve improved catalytic activity by perturbing the reactivity of *maximized* Pt atoms at the nanoparticle-support

interface, which requires the development of support featuring strong electronic interactions with nanoparticles and highly heterogeneous surface structure at the atomic scale. Having this in mind, we identified fullerene (C₆₀) as a potential candidate, since it has been extensively used in hybrid photovoltaics owing to its high electron affinity. Besides, we also found that it is straightforward to prepare C₆₀ bulk crystals, and their highly curved surface and very large lattice distance further provide opportunities in introducing surface confinement effect when anchoring metal nanoparticles. However, there are challenges of using C₆₀ bulk crystal as catalyst substrate, *i.e.*, its low surface area and unsatisfactory conductivity. To overcome these challenges, we developed a facile approach to synthesize a two-dimensional C₆₀ nanosheet (as shown in Fig 1a.), at the meanwhile anchoring metal clusters/nanoparticles onto the uniform surface of C₆₀ nanosheet. Frankly, we have also tried to introduce Cu nanoclusters/nanoparticles onto the thin C₆₀ nanosheet, aiming to develop a new catalyst/substrate composite for electrochemical CO₂ reduction. Unfortunately, we have yet to find a suitable chemical reductant to slow down the growth rate of Cu crystals and induce the controllable C₆₀ bulk crystal exfoliation. Using the same procedure, it actually ended up with a mixture of large Cu particles and the C₆₀ bulk crystals. Thus, only limited substrate effects were obtained when using this mixture for electrochemical CO₂ reduction. Currently, we are trying to decorate the Pt nanocluster on C₆₀ with Cu using underpotential deposition, in this way hopefully we can obtain the desired catalyst composite indirectly, promising preliminary results have been observed.

Taken together, this work offers a two-dimensional carbon support, *i.e.*, C₆₀ nanosheets, which possesses strong electron affinity, heterogeneous surface structure and large lattice distance (surface confinement effect), and could be useful in broader catalytic applications. We have modified the manuscript to demonstrate the above idea more clearly.

Revision made:

Page 2, Line 70

“Buckminsterfullerene (C_{60}) surfaces stand out as promising catalyst-support candidates due to their high electron affinity (each C_{60} molecule can accept up to six electrons)⁴, uniform and undulated structures with large lattice distance. These desired surface properties could enable strong electronic and confinement effects when anchoring metal nanoclusters/particles, and further lead to diverse active sites as we proposed above. Moreover, the strong intermolecular interactions between C_{60} molecules can enable the formation of C_{60} crystals with thin and highly dispersible morphology (*i.e.*, 2D nanosheets),^{5,6} which further enable sufficient electron conductivity⁷⁻¹⁰ and high catalyst-loading-capacity for application as efficient electrocatalyst support. With this design in mind, we developed a facile approach to synthesize two-dimensional C_{60} nanosheets, and constructed a C_{60} /Pt heterostructure (PtC_{60}) as a model system to demonstrate the above catalyst design principle.”

Besides, the diversity of carbon sites is not discussed in the calculation parts. Please give more discussions about the innovation of this work.

Response: We really appreciate this great suggestion. Accordingly, we first calculated the density of states (DOS) of the carbon sites in C_{60} , particularly after the electron transfer occurs from Pt NCs to the carbon sites on C_{60} . As shown in Figure R1 (also Supplementary Fig. 37), the C atoms in Pt/C_{60} contacting with the Pt NCs receive a small but noticeable amount of electron ($DQ = 1.18 e^-$). Consequently, the DOS of these C atoms exhibit a more disperse distribution (Figure R1a) compared to that of the C atoms that are further away from the Pt (Figure R1b). Such dramatic differences in DOS were not observed in the Pt/graphene system. This further confirm the profound electronic interactions at the Pt/C_{60} interface, in contrast to the weak interfacial interactions at the Pt/graphene interface as reported in previous literature.¹¹⁻¹³

Then, we investigated the system of Pt NCs on graphene to extend the understandings on the diversity of carbon sites. As shown in Figure R1c, d, the DOS of graphene C atoms bonded to Pt and not bonded to Pt on graphene are similar, with less dispersion compared to that of the C atoms at the Pt/C_{60} interface, suggesting a much weaker

interfacial interactions agreeing well with the previous literature.^{11–13} We further assessed the improvement of HER activity of Pt/C₆₀ compared to the unsupported Pt NCs and Pt NCs supported on graphene through an electrochemical activity analysis based on a recently developed microkinetic model for alkaline HER on Pt(111).¹⁴ First, the binding energies of H and H₂O species were calculated to vary by as much as 0.45 eV and 0.89 eV among Pt sites at the Pt/C₆₀ interface as compared to 0.17 eV and 0.39 eV for Pt-sites on unsupported Pt NCs, and 0.19 eV and 0.42 eV for sites on Pt/graphene interface, respectively (Fig. 4c, Supplementary Note 7, Table 5 and 6). This profound difference suggests much more heterogeneous properties (in terms of binding strength towards key reaction intermediates) for Pt sites at the Pt/C₆₀ interface compared to the sites on Pt/Graphene. In line with the experimental results (Fig. 3), our analysis based on the microkinetic model shows that Pt sites at the Pt/C₆₀ require much smaller overpotential to achieve a current density of 10 mA cm⁻² compared to the unsupported Pt-sites and the Pt/graphene, respectively (0.09 V and 0.08 V smaller) (Fig. 4e). The activity enhancement becomes more dramatic if we exclude the Pt-sites that are further away from the Pt/C₆₀ interface in the analysis (Supplementary Fig. 41). Overall, the diverse binding properties of the Pt-sites at the Pt/C₆₀ interface strongly improves their alkaline HER activity.

Figure R1. (Supplementary Fig. 37) Maximum, minimum and average value of the electronic densities of states (DOS) per atom for (a) the carbon atoms bonded to the Pt (C-Pt) on Pt/C₆₀, (b) carbon atoms not bonded to Pt on Pt/C₆₀, (c) carbon atoms bonded to the Pt (C-Pt) on graphene, (d) carbon atoms not bonded to Pt on Pt/graphene.

Revision made:

Page 11, line 307

“As shown in Supplementary Fig. 37. the C atoms on C₆₀ contacting with Pt receive a small but noticeable amount of electrons ($\Delta Q = 1.18 e^-$). Consequently, the electronic densities of states (DOS) of these C-atoms exhibit a more disperse distribution (Supplementary Fig. 37a) compared to that of the C-atoms that are further away from the Pt (Supplementary Fig. 37b). Note, such dramatic differences in DOS were not observed in the Pt/graphene system (Supplementary Note 6, Supplementary Fig. 37 c, d), whose structure was similar to the Pd/graphene model designed previously. Thus, the DOS analysis further confirms the profound electronic interactions at the Pt/C₆₀ interface, in contrast to the weak interfacial interactions at the Pt/graphene interface as

reported in the previous literature.^{11–13}”

We further assessed the improvement of HER activity of Pt/C₆₀ compared to the unsupported Pt NCs and Pt NCs supported on graphene through an electrochemical activity analysis based on a recently developed microkinetic model for alkaline HER on Pt(111).¹⁴ First, the differences between the highest and the lowest binding energies of H and H₂O species were calculated to be 0.45 eV and 0.89 eV for Pt sites at the Pt/C₆₀ interface as compared to 0.17 eV and 0.39 eV for Pt-sites on unsupported Pt NCs, and 0.19 eV and 0.42 eV for sites on Pt/graphene interface, respectively (Fig. 4c, Supplementary Note 7, Table 5 and 6). This profound difference suggests much more heterogeneous properties (in terms of binding strength towards key reaction intermediates) for Pt sites at the Pt/C₆₀ interface compared to the sites on Pt/graphene. As a result, some of the Pt/C₆₀ interface sites exhibit low activation energies for Volmer step, whereas other sites on the interface have low activation energies for Heyrovsky and/or Tafel steps (Supplementary Fig. 38, 39, Table 7, 8 and Note 8). After averaging all Pt-sites at the Pt/C₆₀ interface, the rates of the Volmer, Heyrovsky, and Tafel steps on Pt/C₆₀ are calculated to be 94, 21, and 94 times higher than the respective rates on the unsupported Pt NCs, or 90, 15, and 90 times higher than those on Pt/graphene (Fig. 4d). Note that H atoms are calculated to be able to freely move diffusion barriers as low as 0.22 eV) between sites that are highly active in various reaction steps, which could increase HER activity even further (Fig. 4f, Supplementary Fig. 40 and Note 8). In line with the experimental results (Fig. 3), our analysis based on the microkinetic model shows that the Pt/C₆₀ composite requires much smaller overpotential to achieve a current density of 10 mA cm⁻² compared to the unsupported Pt and the Pt/graphene, by 0.09 V and 0.08 V respectively (Fig. 4e). The enhancement becomes much more dramatic if we evaluate normalized activity only for Pt-sites at the Pt/C₆₀ interface (Supplementary Fig. 41). Overall, our model reveals the origin of the improved activity of Pt/C₆₀ towards alkaline HER activity, that is the diversity of binding properties of the Pt-sites at the Pt/C₆₀ interface.

Figure R2. (Fig. 4) **d** The enhancement in the reaction rate constants of the various alkaline HER reaction steps on Pt/C₆₀(011) and Pt/graphene relative to the unsupported Pt NCs estimated through activation energies. **e** LSV of supported Pt/C₆₀, Pt/graphene, and unsupported Pt NCs calculated from the microkinetic model.

Figure R3. (Supplementary Fig. 38) **a**) The geometry of Pt NC adsorbed on C₆₀(011) with the numerated Pt atoms forming the Pt/C₆₀ interface. The fullerene molecules are not displayed for clarity. **b**) The geometry of Pt NC adsorbed on graphene surface.

Figure R4. (Supplementary Fig. 41). Simulated LSV for the 17 interfacial Pt-sites at the Pt/C₆₀ (011) interface, 16 interfacial Pt-sites at the Pt/graphene interface, and all Pt-sites on unsupported Pt NCs normalized per number of Pt atoms.

Supplementary Note 7

“The Gibbs free adsorption energies of these 3 species are also calculated for the Pt sites on Pt/graphene interface and unsupported Pt NCs, and compared with those of the Pt-sites on Pt/C₆₀.”

Supplementary Note 8

“The Volmer step involves the breaking of the bonds in the water molecule to form one adsorbed H and one free OH⁻ species with an electron transfer involved. The adsorbed H species then participate in the subsequent Heyrovsky step for H₂ formation. The thermal water dissociation step is defined as the step to break water into adsorbed H and adsorbed OH species with no electron transfer.¹⁵ It is necessary to define this step for the micro-kinetic modelling, as the model takes into consideration of the OH coverage, which has non-negligible effects on the binding properties of the catalyst. However, the rate of thermal water splitting is negligible compared to Pt-based HER, thus, we believe the Volmer step should be the dominant step for HER in our system.”

“The forward reaction rate constants shed light on the enhancement of the alkaline HER rates in the presence of the C₆₀ support. Specifically, the Volmer step is enhanced by 94 and 90 times, the Heyrovsky step by 21 and 15 times, the Tafel step by 94 and 90 times, and the thermal water dissociation by 9 and 8 times for the 17 interfacial Pt-sites on Pt/C₆₀ as compared to the unsupported Pt NCs and the interfacial Pt-sites on Pt/graphene (Supplementary Table 9), respectively.”

“Similar results are observed when all the surface Pt sites are considered on the Pt/graphene, yielding 0.08 V higher overpotential compared to the activity of the averaged Pt sites of Pt/C₆₀(011) at the current density of 10 mA cm⁻².”

Supplementary Table 5-10

2a. In Fig.2 b and c, the authors used the term ‘in-situ’ to introduce the XPS and UPS spectra. Since the deposition and X-ray measurement is not processed at the same time, thus it is inappropriate (regardless of whether to keep the sample in the high vacuum or not).

Response: We appreciate this suggestion from the referee. We fully agree with the referee and have update the language accordingly, in both the manuscript and the supplementary information.

Revision made:

The caption of Fig. 2

“**Fig. 2 Electronic Characterization and Analysis.** **a** High-resolution Pt 4f XPS spectra of PtC₆₀ and Pt NCs. **b, c** All-in-vacuum XPS and UPS spectra of Pt/C₆₀ film within the same vacuum chamber.”

Page 7, line 181

“To prove the presence of interfacial charge transfer, we conducted all-in-vacuum XPS and ultraviolet photoelectron spectroscopy (UPS) measurements by keeping the deposition of C₆₀ thin-film and Pt layer within the same vacuum chamber,

(Supplementary Fig. 18)”

Page 15, line 419

“The all-in-vacuum UPS and XPS measurements (including the Pt atomic layer deposition and C₆₀ thin film deposition)”

Supplementary Fig. 16. Schematic illustration of the home-built XPS/UPS.

2b. Furthermore, using thickness to distinguish different samples is not a right way; the ‘Pt loading’ is better. In fact, the Pt has an atomic radius of 0.139 nm.

Response: We thank the referee for this great comment. We fully agree that using “Pt loading” is much more accurate to describe the samples for the XPS/UPS test. In the XPS/UPS measurement, the Pt loading was estimated by measuring the attenuation of C 1s peak before and after Pt deposition and then calibrated by a quartz crystal microbalance (QCM) located in front of the sample stage. We have replaced the “Pt thickness” with “Mass loading of platinum” in the manuscript accordingly.

Revision made:

Page 7, line 185

“(at the loading of 8.6 ng cm⁻²)”

Page 7, line 188

“when the loading of the deposited Pt reaches 1.1 μg cm⁻²”

Page 15, line 425

“The mass loading of the Pt was estimated by measuring the attenuation...”

Figure R5. (Fig. 2 b-d) Fig. 2 Electronic Characterization and analysis. b, c All-in-vacuum XPS and UPS spectra of Pt/C₆₀ film within the same vacuum chamber. **d** Binding energy of Pt 4f_{7/2} and work function depending on the amount of Pt deposited on C₆₀ film corresponding to **b** and **c**.

2c. Samples with thickness from 0.004 to 0.06 nm are not even achieve one single layer of Pt atom, which means they only reflect the coverage of Pt atom on C₆₀ film. Therefore, the negative shift of binding energy with the increase of Pt loading is due to the higher loading of metal Pt⁰ sharing the electron transfer from the C₆₀ film. More detailed, for the 0.004 nm sample, its binding energy is ~72 eV, which is still lower than that of Pt²⁺ (72.5 V). Therefore, it could be more persuasive to speculate that the high valence is formed during the synthesis process, where some of the Pt_x⁺ can be stabilized by the C₆₀ due to the support effect. Please give some explanations.

Response: We thank the referee for this important comment. We completely agree with the referee that the combined phenomena may also be resulted from the stabilization of the Pt^{δ+} by the C₆₀ substrate. Our original intention of conducting these all-in-vacuum UPS/XPS measurements is to prevent the oxidation of Pt by air, at least to the best extent (~10⁻¹⁰ mbar in our customized ultrahigh vacuum chamber). To provide more precise conclusions for this matter, we have conducted the following analysis and

discussions:

First, we have carefully revisited our XPS data, especially for the O 1s peaks for Pt NCs and Pt/C₆₀, respectively. The binding energy shift/peak deconvolution of the O 1s can help us understand the valence state change of Pt induced by oxidation. Generally, XPS spectrum of O 1s of oxygen adsorbed on Pt reported in literature have a peak from 528.5 eV to 530.7 eV.¹⁶ In this work, the O 1s peak of Pt NCs XPS spectrum centered at ~531.0 eV (Figure R6 bottom). Hence, if the C₆₀ nanosheets stabilizes the oxidized Pt and adsorbed oxygen on Pt, the binding energy of O 1s of PtC₆₀ would shift towards a smaller binding energy region when comparing to the non-supported Pt NCs. However, in contrast, the O 1s binding energy shift oppositely after the introduction of C₆₀ support (Figure R6 top). Both of the fitted O 1s spectra of PtC₆₀ and Pt NCs display a peak at ~530.7, which can be attributed to the Pt-O bond formed by the oxidation of Pt nanoclusters surface. More importantly, this sub-peak, and even the entire O 1s peak did not increase (with regard to the peak intensity of Pt-O) after the introduction of C₆₀. On the other hand, the positive shift of O 1s peak of PtC₆₀ can be assigned to the increase of oxygen species (*i.e.*, adsorbed H₂O, or OH) on C₆₀. Overall, from the analysis of O 1s XPS spectra, it is more likely that C₆₀ substrate induce the charge transfer from Pt to the Pt/C₆₀ interface which caused the Pt 4d peak shift.

Accordingly, we reanalyze the XPS spectra of Pt 4f (Fig. 3a). We believe it is more appropriate to deconvolute the previous Pt^{δ+} peak into Pt²⁺ and a separate Pt^{δ+} (0<δ<2) species, which are resulted from the surface Pt oxide and Pt^{δ+} species induced by the charge transfer at the Pt/C₆₀ interface, respectively.

Revision made:

Figure R6. (Supplementary Fig. 15) b) O 1s of PtC₆₀, Pt NCs, and C₆₀ precursor powder.

Figure R7. (Fig. 2) a) High-resolution Pt 4f XPS spectra of PtC₆₀ and Pt NCs.

“The high-resolution XPS Pt 4f signals for Pt NCs can be deconvoluted into two components, corresponding to metallic Pt⁰ and oxidized Pt²⁺ species, at 71.2/74.5 eV and 72.6/76.3 eV, respectively (Fig. 2a).^{17–19} However, the broader Pt 4f peaks of PtC₆₀ compared to those of the Pt NCs suggest an extra content of Pt^{δ+} species. To explore the origin of the increased Pt^{δ+} species, *e.g.*, whether the C₆₀ substrate stabilizes the oxidized form of Pt, we conducted curve fittings for the associated O 1s species for both Pt NCs, PtC₆₀ and C₆₀ precursor powder (Supplementary Fig. 15). It turns out that we could not find an increase of O 1s that is associated to the Pt oxide. Thus, based on the computational results discussed later, we tentatively attribute the extra content of Pt^{δ+} species in PtC₆₀ to the interfacial electron transfer from Pt to C₆₀. Moreover, we have prepared additional control catalysts: commercial Pt/C (20% Pt loading) and Pt deposited on XC-72 carbon black (Pt/CB) (40% Pt loading)). The Pt 4f XPS of Pt/CB exhibits sharp peaks and peak compositions similar to those of the Pt NCs, suggesting that the electronic interactions between Pt and graphitic carbon are negligible (Supplementary Fig. 16). In the following, we will focus only on the analysis of Pt/C catalysts because they feature similar intrinsic activity and HER mechanism as Pt/CB according to the Tafel plots (Supplementary Fig. 15).”

(3) The fitting procedure for UPS spectra should be introduced in the manuscript.

Response: We appreciate this great suggestion from the referee. Accordingly, we have provided the fitting procedure for UPS spectra in both the Method section in the manuscript and Supplementary information (Supplementary Fig. 43).

Revision made:

Method

“The all-in-vacuum UPS and XPS measurements (including the Pt atomic layer

deposition and C₆₀ thin film deposition) were all performed in our customized ultrahigh vacuum chamber ($\sim 10^{-10}$ mbar), using He I (21.2 eV) and Mg K α (1253.6 eV) as excitation sources, respectively. Using a sample bias voltage of -7 V, the work function of the sample was obtained by the secondary electron cut-off in the low kinetic energy region using the following eq. 2.^{20,21} (Supplementary Fig. 43a) The Fermi level was calibrated with respect to the sputter-cleaned gold foil measured at room temperature. The mass loading of the Pt was estimated by measuring the attenuation of C 1s peak before and after Pt deposition and then calibrated by a quartz crystal microbalance (QCM) located in front of the sample stage.

$$\phi = h\nu - W \quad [1]$$

$$\phi = h\nu - (E_{SECO} - E_F) \quad [2]$$

W is the spectrum from E_F to E_{SECO} . $h\nu$ is 21.21 eV (He). E_F is 0 eV, E_{SECO} is 15.7 eV, 15.6 eV, 15.5 eV, 15.3 eV, and 14.9 eV for 0 nm, 0.004 nm, 0.06 nm, 0.02 nm, and 0.5 nm Pt average thickness (Supplementary Fig. 43b). Hence, the work function calculated *via* eq. 2 is 5.5 eV, 5.6 eV, 5.7 eV, 5.9 eV, and 6.3 eV, respectively.”

Figure R8 (Supplementary Fig. 43) a) UPS spectra and b) high-resolution UPS spectra during Pt deposition on C₆₀ film

(4) Considering that it is not easy to control the extreme low loading of catalysts, the author should provide the error bars for all the electrochemical performance in Fig. 4b, c.

Response: We thank the referee for this great suggestion. Accordingly, we have provided the error bars for the electrochemical performance test on RDE with different catalysts loadings. The related Figures are revised as well. (Sample size $n = 3$; data presented as mean \pm one standard deviation)

Revision made:

Figure R9. (Fig. 3) Fig. 3 Alkaline HER. a LSV (linear sweep voltammetry) of Pt catalysts with the loading of 0.4 mg cm^{-2} and bare C_{60} . **b** Overpotentials of the catalysts at current densities of 10, 50, 150 mA cm^{-2} in **a**. **c** LSV of the catalysts with very low catalyst loading of 0.004 mg cm^{-2} for assessing the intrinsic activities. Inset: geometric current density of the samples at an overpotential of 200 mV. **d** TOFs of the catalysts with the catalyst loading of 0.004 mg cm^{-2} . **e** Corresponding Tafel plots obtained from LSV in **c**.

Figure R10. (Supplementary Fig. 27) LSV curves of a) PtC₆₀, b) Pt/C, c) Pt NCs normalized by geometry area of electrodes with different catalyst loadings in 1.0 M KOH.

Figure R11 (Supplementary Fig. 28) LSV curves normalized by Pt mass loading with different catalyst loadings of a) 0.4 mg cm⁻², 0.04 mg cm⁻², and c) 0.004 mg cm⁻² in 1.0 M KOH.

Figure R12. (Supplementary Fig. 30) LSV curves normalized by ECSA with different catalyst loadings of a) 0.4 mg cm⁻², 0.04 mg cm⁻², and c) 0.004 mg cm⁻² in 1.0 M KOH.

Page 8, line 220

“The alkaline HER on PtC₆₀ was studied using a rotating disk electrode in 1.0 M KOH. PtC₆₀ with modest loading of ~0.4 mg cm⁻² reaches the current densities of 10 mA cm⁻², 50 mA cm⁻² and 150 mA cm⁻² at overpotentials of 24.3 mV, 53.2 mV and 110.0 mV, respectively.”

Page 9, line 248

“As shown in Fig. 3c, with a low catalyst loading of 0.004 mg cm⁻², PtC₆₀ exhibits a much higher current density compared to those of Pt/C and Pt NCs at all potentials. For instance, PtC₆₀ affords a geometric current density of 2.3 mA cm⁻² at the overpotential of 100 mV, which is 11 and 7 times larger than those of Pt/C (0.19 mA cm⁻²) and Pt NCs (0.34 mA cm⁻²). Similar activity trends were observed on electrodes with various catalyst loadings before and after normalization by the Pt mass loading (Supplementary Fig. 27 and 28). Under the same conditions, the mass activity of PtC₆₀ is estimated to be 1.55 A mg⁻¹, which is much higher than those of the Pt/C (0.24 A mg⁻¹) and Pt NCs (0.08 A mg⁻¹) at overpotential of 100 mV. Moreover, the HER activity of PtC₆₀ also exceeds those of Pt/C and Pt NCs after the normalization by the electrode surface area estimated *via* under potential deposited H (Supplementary Fig. 29, 30 and Table 3), which was shown to reflect the specific active surface area of Pt-based materials.^{22,23} Specifically, the turnover frequency (TOF) of PtC₆₀ (17.6 s⁻¹) estimated based on the number of surface Pt-sites was remarkably higher than those of Pt/C (1.5 s⁻¹) and Pt NCs (2.8 s⁻¹) (Fig. 3d) at overpotential of 100 mV.”

(5) Anion exchange membrane is the solid ‘polymer’ electrolyte. For AEM water electrolysis, the chronopotentiometry (CP) method is more common and has more advantages in precisely evaluating performance than the linear sweep voltammetry (LSV) method. The cell performance measured by LSV with a fast scan rate would be limited by the relative slow mass transfer process, where the fast voltage change doesn’t allow the reacting system being stable, thus showing an inaccurate and less

reproducible current-voltage real-time response. By applying multiple CPs to record the stable current-voltage response, the polarization curve of the cell can be achieved (*Adv. Mater.* 2022, 34, 2203033). Moreover, the iR compensation is unnecessary for the AEMWE testing because the resistance would change with the applied current density. As a result, the fixed resistance to plot the linear ohmic loss in Fig. 5d-f might be imprecise.

Response: We really appreciate the great suggestions and guidance from the referee. Accordingly, we have remeasured the performance of AEM based on the procedure reported from the recent work of *Adv. Mater.* 2022, 34, 2203033 (reference 78 in the manuscript).²⁴ The new results on the AEM performance are presented in the revised manuscript accordingly. The overall performance trends among the sample of interests remain the same.

Revision made:

Figure R13 (Fig. 5) Performance of AEM electrolyser. a LSV curves of the AEM electrolyzers using PtC₆₀||IrO₂, PtC||IrO₂ and Pt NPs||IrO₂, respectively.

Page 12, line 345

“Specifically, the electrolyser using PtC₆₀ reaches a current density of 1.0 A cm⁻² at a cell voltage of 2.01 V, which is much lower than those of the electrolysers with Pt/C (2.18 V) and Pt NCs (2.58 V) catalysts (Fig. 5a, 5b). Consequently, the energy efficiency of the PtC₆₀-based AEM electrolyser reaches 74% at 1 A cm⁻² (Supplementary Table 11, Note 9), surpassing those of Pt/C (67.8%) and Pt NCs (57.3%) containing electrolysers (Fig. 5b).²⁵”

Page 18, line 523

“Prior to AEM testing, 10 cycles of cyclic voltammetry (CV) were conducted from 0 V to 1.5 V of cell voltage at a scan rate of 50 mV s⁻¹. Then the AEM electrolyser was conducted at 2 mA cm⁻² for 5 minutes to stabilize. Subsequently, the electrolyser was tested at 2 mA cm⁻², 10 mA cm⁻², 20 mA cm⁻², 50 mA cm⁻², 100 mA cm⁻², 200 mA cm⁻², and increased in 200 mA cm⁻² steps until reaching 2 A cm⁻² via a chronopotentiometry method, and the potential was recorded, measuring the potential for 15 s at each step to collect the current-voltage curve. For the stability measurement, the catalyst ink was prepared in the same way as above. Then, 3.0 mg of PtC₆₀ and 3 mg of Ir/C were deposited onto the nickel foams employed as cathode and anode with an active surface area of 1 × 1 cm², respectively. The CP test was carried out at 60 °C, using 1.0 M KOH as the electrolyte with a flowing rate of 40 mL/min.”

(7) At 60°C, the thermodynamic potential of water splitting is not 1.23 V anymore. Please check the overpotential listed in the Fig. 5b. Again, error bar should be provided.

Response: We thank the referee for this important point. Accordingly, we have estimated the thermodynamic potential of water splitting based on the Nernst equation, which is about ~1.20 V ($p = 101.3$ kPa) at 60 °C.²⁶ In addition, we plotted both the polarization curves and the energy efficiency on the function of “Cell Voltage” instead of “Overpotential”, which is a common practice in the related work.^{24,27,28} In the revised

version of manuscript, we present the performance of AEM electrolyser with “Cell voltage” as the y-axis. Accordingly, we have updated the electrolyser energy efficiencies based on this updated value. At least three repeats were conducted to provide the error bars, as shown in Fig. 5a and b.

Revision made:

Figure R14 (Fig. 5) b Cell voltage at the current density of 1 A cm⁻² and the corresponding energy efficiency. (Sample size n = 3; data of **a b** presented as mean ± one standard deviation)

Page 12, line 345

“Specifically, the electrolyser using PtC₆₀ reaches a current density of 1.0 A cm⁻² at a cell voltage of 2.01 V, which is much lower than those of the electrolysers with Pt/C (2.18 V) and Pt NCs (2.58 V) catalysts (Fig. 5a, 5b). Consequently, the energy efficiency of the PtC₆₀-based AEM electrolyser reaches 74% at 1 A cm⁻² (Supplementary Table 11, Note 9), surpassing those of Pt/C (67.8%) and Pt NCs (57.3%) containing electrolysers (Fig. 5b).²⁵”

Response to Reviewer 2:

This manuscript reports a new catalyst composed of Pt nanoclusters anchored on C60 sheets for enhanced HER performance. The methods are sound and the analyses and results are representing the cutting-edge research. In particular, the mechanistic investigation helps to understand the obtained performance and the industrial devices demonstrate the commercialisation potential. I would recommend publications after addressing the following questions:

We thank the referee for all the important and constructive comments/suggestions, which help us a lot in improving the quality of this manuscript. Corresponding revisions are made and described below in a point-by-point fashion. All changes have been highlighted in the revised manuscript and supplementary information files.

1. The main challenge in alkaline membrane water electrolysis is the durability and ion conductivity of the membrane. The authors need to justify the relevance of catalyst development in this broad research context.

Response: We thank the referee for raising this important point. We completely agree with the referee, the major challenge of the practical implementation of AEM water electrolyser lies on the durability and ion conductivity of the anion exchange membrane. Thus, tremendous research efforts have been spent on the development of AEMs with the above-mentioned desired performance, and promising AEM candidates have been reported.^{29,30} On the other hand, many TEA (Techno-Economic Analysis) analysis suggest that the electricity cost remains the major cost for the overall hydrogen production.³¹ Considering that the kinetics of the HER in alkaline media are typically one to a few orders of magnitude lower than the acidic HER, even for the state-of-the-art Pt catalysts. Therefore, it remains desirable to continue the search for more efficient alkaline HER catalyst, so that we can further reduce the production cost for green hydrogen, which is especially important for hydrogen implementation at large scale.

Revision made:

Page 1, line 37

“While the unsatisfactory durability and ion conductivity of the alkaline exchange membrane still hinder the practical implementation of AEM water electrolyser, there are promising candidates reported recently which may lead to breakthroughs for this technology.^{9,10} Thus, it remains desirable to develop efficient alkaline HER catalyst, since the electricity cost (large overpotential required causing high energy demand) is recognized to dominate the overall green hydrogen production cost.^{9,11}”

2. While the activity of HER catalyst is a problem in alkaline electrolysis, the use of transition metals rather than Pt is often applied due to the relatively good stability and low cost of transition metals in alkaline media. The authors thus need to justify why Pt is of interest in this study.

Response: We thank the referee for this important comment. We fully agree that great efforts have been spent on developing cost-effective transition metals based alkaline HER catalysts,^{32,33} and we also believe they can be very promising for future AEM applications. However, at present, platinum-based materials remain the state-of-the-art alkaline HER catalyst. Hence it is still worth promoting the activity and stability of Pt for HER. On the other hand, our project mainly focuses on the development of two-dimensional catalyst support and the interface engineering between catalytic material and this two-dimensional C₆₀ support, with the aim of introducing the design principle of diverse active-site for multi-step electrochemical reactions. We think choosing the classic Pt-based catalysts would best elaborate our design as well as getting more attentions from the community due to the comprehensive researches on Pt-based catalysts. In the future, we would love to extend our design to earth-abundant transition metal-based materials and other related electrochemical application, *e.g.*, CO₂ electroreduction. We have revised the manuscript to explain this view in the introduction section.

Revision:

Page 2, line 67

“Moreover, we believe this strategy can be applied broadly to the design of other catalysts, including earth-abundant transition metal HER catalysts to further reduce the cost of AEM devices.”

3. P3, what is the difference between the Volmer step and the water dissociation step in alkaline HER?

Response: We thank the referee for this great question. Indeed, it is not common to discuss water dissociation step in HER. Generally, the first electrochemical step of alkaline HER is considered to be: water dissociation to generate adsorbed hydrogen (*H) and hydroxide ion (OH⁻) through the Volmer step.¹⁵ In this work, we also considered thermal or chemical water dissociation into adsorbed *H and *OH, which provides a pathway for the buildup of *OH species on the catalyst surface. The recent work by Chan *et al.* established a microkinetic model for studying HER under alkaline condition, and concluded that the *OH coverage and desorption plays important role in the entire catalytic process.¹⁴ Thus, we included thermal “water-dissociation step”, and conducted the same microkinetic analysis (Fig. 4e) to understand the activity of Pt/C₆₀ interface in alkaline HER. Ultimately, our calculations showed that this step is not important for the HER mechanism on Pt/C₆₀. To avoid confusion, we have removed the bar of water dissociation step from Fig. 4d, as it does not contribute to the rate of alkaline HER.

Revision made:

Figure R15. (Fig. 4d) d The enhancement in the reaction rate constants of the various alkaline HER reaction steps on Pt/C₆₀(011) and Pt/graphene relative to the unsupported Pt NCs estimated through activation energies.

Page 12, line 327

“After averaging all Pt-sites at the Pt/C₆₀ interface, the rates of the Volmer, Heyrovsky, and Tafel steps on Pt/C₆₀ are calculated to be 94, 21, and 94times higher than the respective rates on the unsupported Pt NCs, or 90, 15, and 90 times higher than those on Pt/graphene (Fig. 4d).”

Supplementary Note 8

“The Volmer step involves the breaking of the bonds in the water molecule to form one adsorbed H and one free OH⁻ species with an electron transfer involved. The adsorbed H species then participate in the subsequent Heyrovsky step for H₂ formation. The thermal water dissociation step is defined as the step to break water into adsorbed H and adsorbed OH species with no electron transfer.¹⁸ It is necessary to define this step for the micro-kinetic modelling, as the model takes into consideration of the OH coverage, which has non-negligible effects on the binding properties of the catalyst. However, the rate of thermal water splitting is negligible compared to Pt-based HER, thus, we believe the Volmer step should be the dominant step for HER in our system.”

4. P3, more details are needed on the C₆₀ crystals, e.g., what is the interaction that holds the C₆₀ together into a flat 2D sheet rather than collapsing into agglomerates?

Response: We really appreciate this important question from the referee. The two-dimensional C₆₀ nanosheets in this work were synthesized *via* a modified liquid-liquid interface synthetic method.³⁴ Other strategies, *i.e.*, organic cation slicing strategy,³⁵ were also capable of preparing two-dimensional C₆₀ materials. With regard of the formation mechanism of C₆₀ crystals, Rao *et al.* previously suggested that individual C₆₀ molecules can be covalently linked together to form polymeric structures.⁵ Similarly, Blank *et al.* also reported that polymerization between C₆₀ occurs *via* intermolecular covalent bonds to produce a layered structure of C₆₀ under elevated pressure.⁶ Recently, Hou *et al.* prepared a single monolayer quasi-hexagonal-phase C₆₀ using an organic cation slicing strategy,³⁵ and observed the identifiable bridge bond between each spherical C₆₀ cages. In our case, we also observed a strong peak at 1400-1500 cm⁻¹ reflects the polymerization of C₆₀ as shown in Figure R16 (Supplementary Fig. 6). Compared with the C₆₀ powder, the weakened peak at 1430 cm⁻¹ of PtC₆₀ further confirmed the strengthened intermolecular interaction between the C₆₀ molecules.^{36,37} Overall, on the basis of previous research and the FT-IR spectra, we believe the intermolecular bonds between C₆₀ clusters are responsible for the formation of the 2D nanosheets, preventing the entire structure from collapsing into agglomerates. In fact, we believe these intermolecular bonds are quite strong as evidenced based on the thermal stability experiments (Supplementary Fig. 9).

Figure R16. (Supplementary Fig. 6) Fourier transform infrared (FT-IR) spectra of PtC₆₀, C₆₀ bulk crystal, C₆₀ powder precursor, and Pt NCs.

Revision made:

Page 3, line 74

“Moreover, the strong intermolecular interactions between C₆₀ molecules can enable the formation of C₆₀ crystals with thin and highly dispersible morphology (*i.e.*, 2D nanosheets),^{41,42} which further enable sufficient electron conductivity^{43–46} and high catalyst-loading-capacity for application as efficient electrocatalyst support. With this design in mind, we developed a facile approach to synthesize two-dimensional C₆₀ nanosheets, and constructed a C₆₀/Pt heterostructure (PtC₆₀) as a model system to demonstrate the above catalyst design principle.”

Page 5, line 121

“The FT-IR spectrum of PtC₆₀ (Supplementary Fig. 6) exhibits weakened peaks at 1175 cm⁻¹ and 1430 cm⁻¹ compared with that of the C₆₀ precursor, suggesting enhanced intermolecular interactions among C₆₀ molecules within PtC₆₀.^{50,51}”

In addition, as the exfoliation is caused by vigorous chemical reaction, how uniform could the sheets be?

Response: We thank the referee for this important question. Indeed, nanosheets synthesized *via* exfoliations may exhibit diverse shape, sizes and even thicknesses. Besides, these nanosheets are usually easy to stack together due to the *Van-der-Waals forces*. However, based on the atomic force microscope (AFM) measurements (Figure R17), we found that the obtained 2D PtC₆₀ sheets are relatively uniform with minor variations in both thickness (5-10 nm) and sizes (a few μm). Overall, the thin nature of the C₆₀ nanosheets will enable the exposure of Pt active sites and promote the catalytic activity towards HER. Accordingly, we have provided the AFM image showing the nanostructure of PtC₆₀ in supplementary information (Supplementary Fig. 3).

Revision made:

Figure R17. (Supplementary Fig. 3) a) AFM image of PtC₆₀, b) depth profile of a cross-section corresponding to the line in the a).

Page 5, line 118

“AFM (Fig. 1c, Supplementary Fig. 3) and TEM images (Supplementary Fig. 4) suggest that the size of PtC₆₀ nanosheets ranges from 200 to 1000 nm with a thickness of ~5 nm. The profound Tyndall effect (Supplementary Fig. 5) further confirms the thin nature of the PtC₆₀ nanosheets.”

5. P4, what is the origin of the strong confinement effect in regulating the narrow size distribution (2 nm) of Pt clusters?

Response:

We thank the referee for this great question. We believe that two possible reasons lead to the strong confinement effect in regulating the narrow size distribution of Pt clusters. First, the unique surface morphology of the 2D-C₆₀ nanosheets, on which the curved nature of C₆₀ together with the unusually large lattice distance leads to the formation the repeating and undulating valleys with regular sizes on the support surface. These surface pores will help anchoring the Pt cluster on the surface. Second, we also observed a strong electronic interaction between the Pt-atoms and the C₆₀ interface (*i.e.*, XPS, UPS and XAS), which may also play a role in stabilizing the Pt-clusters on the C₆₀ surface. The electronic interactions were further explored in the DFT analysis section, and is believed to be the origin of the diverse Pt-sites at the Pt/C₆₀ interface. Accordingly, we have provided additional discussion of the confinement effects in the manuscript.

Revision made:

Page 8, line 205

“Taken together, the strong confinement effect of C₆₀ nanosheets leads to the formation of the Pt clusters with narrow size distribution (2 nm), which is likely originated from the combination of the unique surface morphology of C₆₀ nanosheets and the electronic interactions between the Pt atoms and the C₆₀ at the interface.”

6. P6, I would suggest to move the WY contour in Fig. S17 to the main text. They provide clear information on the bonding states among different samples.

Response: We appreciate this suggestion, and have moved the wavelet transform contour to Fig. 2 in the manuscript accordingly.

Revision made:

Page 8, line 200

“In addition, the extended X-ray adsorption fine structure (EXAFS) (Pt L₃-edge, Fig. 2f and Supplementary Fig. 19) and the corresponding wavelet transform (Fig. 2h, i and Supplementary Fig. 20).”

Figure R18. (Fig. 2) h and i Wavelet transform for the k^3 -weighted EXAFS spectra for PtC₆₀ and Pt NCs, respectively.

7. P8, more details are needed to explain the mass transfer effect as shown in Fig. S25.

Response: We thank the referee for this great suggestion. Accordingly, we have added more discussions to the manuscript and the Supplementary Information with regard the mass transportation effect.

Revision made:

Supplementary Note 2:

“Sluggish mass transport on electrode surface is an important issue to consider during investigations of the intrinsic activity of Pt-based catalysts toward HER. Even under RDE experiment, the H₂ evolution is so fast and the measured current can be easily limited by mass transport of H₂ away from the electrode.^{38,39} The model of Hansen et al. displays that the intrinsic activity of Pt under acidic medium on RDE system has no effect on the catalytic performance. The frequently reported Tafel slope of 30 mV/dec for HER is just the apparent Tafel slope of the diffusion controlled overpotential. Thus, mass-transport limitations prevent a genuine comparison of intrinsic activities. In this work, we adapted the strategy of decreasing the catalyst loading to mitigate the influence of mass-transport effects in evaluation of catalysts.”

“Based on the above model, the i_l is calculated to be 0.0316 A. The i_0 is estimated from polarization curves as indicated in Supplementary Table 2. As shown in Supplementary Fig. 26a, the facile electrode kinetic of PtC₆₀ is limited by the diffusion of produced H₂ when a typical catalyst loading was used. Due to the Nernstian limitation, the intrinsic kinetics plays a minor role under these catalyst loadings with large exchange currents for all the samples, in which i/i_l at -0.3 V vs. RHE is -0.95, -0.88, and -0.87 for PtC₆₀, Pt/C, and Pt NCs (Supplementary Fig. 26d), respectively. As a result, all polarization curves stack together showing a small discrepancy of the HER activity. However, the interval of the corresponding polarization curves increases along with the decrease in catalyst loading (Supplementary Fig.26b-c). At the loading of 0.004 mg cm⁻², the i/i_l of PtC₆₀ at -0.3 V vs. RHE is 0.3, which is ~15 times of Pt/C, and ~24 times of Pt NCs (Supplementary Fig. 26d). In this case, the slower overall HER production rate at low catalyst loading mitigates the influence of mass-transfer effect of H₂, so that profound differences in HER kinetics can be observed among these Pt catalysts with different intrinsic activities. ”

Figure R19. (Supplementary Fig. 26) Mass transportation effect: simulated polarization curves of PtC₆₀, Pt/C, and Pt NCs with different loading amounts of a) 0.4 mg cm⁻², b) 0.04 mg cm⁻², and c) 0.004 mg cm⁻² catalysts. d) ratio of i to i_L with different catalyst loading of PtC₆₀, Pt/C, and Pt NCs at -0.3 V vs. RHE.

8. P10, is it the binding energy of H or OH in Fig. 4c?

Response: We appreciate the careful review, and apologize for the confusion. In fact, Fig. 4c presents the overall distributions of the binding Gibbs free energies of H (top), OH (middle), and H₂O (bottom) on unsupported Pt₁₄₀ and Pt₁₄₀/C₆₀ under the same conditions, respectively. We have revised the caption of Fig. 4c to make it clear.

Revision made:

Figure R20. (Fig. 4) c Distribution of binding Gibbs free energies of H (top), OH (middle), and H₂O (bottom) on unsupported Pt NCs compared to the much wider distributions calculated on Pt/C₆₀.

Response to Reviewer 3:

A contemporary challenge in the electrocatalytic hydrogen production is the development of catalysts for accelerating alkaline HER by improving the intrinsic activity of Pt. This is where the current work contributes by constructing Pt-sites at Pt/C60 interface with diverse binding properties at key-reaction intermediates of HER and achieving enhanced activity. Thus, the work is novel and certainly can be considered for publication in a prime journal. However, the following issues require the attention of the authors to clarify vague points of the manuscript and enhance its quality:

We are very grateful for the in-depth and constructive comments, suggestions from the referee. Corresponding revisions are made and described below in a point-by-point fashion. All changes have been highlighted in the revised manuscript and support information files.

(1) What is the thickness of the bulk C₆₀ crystals, obtained via the liquid-liquid interfacial precipitation method?

Response: We thank the referee for this important question. Indeed, it is not obvious to tell the thickness of the bulk C₆₀ crystals based on the TEM images (Figure R14a). However, based on the AFM image (Figure. R14 e. f), we believe the thickness of C₆₀ bulk crystal is about 20 nm, and no obvious aggregations were observed from both the TEM and AFM images.

Revision made:

Page 4, line 105

“As shown in the transmission electron microscopy (TEM, Supplementary Fig. 1a-b), the obtained C₆₀ crystals exhibit sizes ranging from 200 nm to 1 μm, much smaller compared to C₆₀ crystals reported elsewhere.^{45,47–49} The TEM and Fast Fourier transform (FFT, Supplementary Fig. 1c-d) images indicate that the C₆₀ crystals display high crystallinity along with the (1 $\bar{1}$ 0) basal plane with markedly irregular edges and

an average thickness of ~20 nm (Atomic Force Microscope (AFM) images in Supplementary Fig. 1e-f), which are beneficial for the subsequent exfoliation step.”

Figure. R21 (Supplementary Fig. 1). a-c) TEM images of the C₆₀ bulk crystal, d) the FFT pattern derived from the selected area in (c), e) typical AFM image of C₆₀ bulk crystal, f) depth profile of a cross-section corresponding to the line in the e).

(2) It is not clear how C₆₀ ended up carrying oxygenated species. What is the purity of commercially available C₆₀ used and what are its spectroscopic characteristics based on IR and XPS? This information is important and it is needed for comparison purposes with the C₆₀ bulk crystals and PtC₆₀, showing the presence of carboxylic acid and hydroxyl moieties.

Overall, major revisions are necessary and only if the authors sufficiently address the aforementioned points, the work can be recommended for publication.

Response: We thank the referee for raising this very important point. We completely agree with the referee on this. It is very strange that C₆₀ ended up carrying oxygenated species, e.g., carboxylic groups, especially, the PtC₆₀ composite is synthesized within a

reducing environment. To understand this, we first measured the IR of the as purchased C₆₀ powder (99.0% from TanFeng. Int.). As expected, we could not observe any significant oxygenated species, as shown in Fig. R23. Furthermore, we also carried out XPS analysis on the as purchased C₆₀ powder. As shown Figure R22a, only a sharp peak of C-C and C=C can be observed in the C 1s peak, and only minor adsorbed oxygen and hydroxyl species can be observed in the O 1s peak, suggesting that no quantifiable amount of oxygenated functional groups exist within the C₆₀ powder before the liquid-liquid precipitation process.

Then, we retested the synthesized C₆₀ bulk crystals for IR. However, to avoid the potential contaminations (*i.e.*, residual solvents, moisture (severe issue in Singapore), *etc.*) on the surface of the sample, all samples have been dried in a vacuum oven for at least 24 h, which was not done in our previous experiments. As shown in Figure R23, there are no significant changes in the IR spectrum of the C₆₀ bulk crystals compare to that of the C₆₀ powder precursor, except the obvious decrease of the signature peak at ~ 1200 and ~ 1430 cm⁻¹, which is likely induced by the intermolecular interactions between the lattice C₆₀ molecules.^{36,37}

Finally, we also re-measured the IR spectrum of PtC₆₀ after thoroughly drying process. As a result, most of the previous major peaks of the oxygenated species (C-O, and O-H) disappeared, indicating that the contamination issues indeed misled the previous IR analysis. However, there is still a minor peak at around 1100 cm⁻¹, which we tentatively attribute to the Pt/C₆₀ interface. Besides, we also cannot observe any obvious oxygenated groups based on the O 1s XPS spectrum of PtC₆₀ (Figure R6). Nonetheless, we believe these minor oxygenated species likely will not exist and/or influence the catalysis since that the Pt is the active-site, which will remain as metallic phase based on its Pourbaix diagram.⁴⁰

Accordingly, we have removed the related discussions on the hydrophilicity, which was linked to the oxygenated group previously. With regard of the enhanced hydrophilicity of PtC₆₀, we currently do not have a clear understanding, however, we suppose it is

associated with the very thin nature of the PtC₆₀ composite, as well as the relatively high loading of the Pt on the C₆₀ nanosheet which may change its physical properties.

Figure. R22 High-resolution XPS of C₆₀ powder a) C 1s of C₆₀ powder, and b) O 1s.

Revision made:

Figure R23. (Supplementary Fig. 6) Fourier transform infrared (FT-IR) spectra of PtC₆₀, C₆₀ bulk crystal, C₆₀ powder, and Pt NCs.

Figure R24. (Supplementary Fig. 15) O 1s of PtC₆₀, Pt NCs and C₆₀ precursor powder.

Page 7, line 165

“To explore the origin of the increased Pt^{δ+} species, *e.g.*, whether the C₆₀ substrate stabilize the oxidized form of Pt, we conducted curve fittings for the associated O 1s species for both Pt NCs, PtC₆₀ and C₆₀ precursor powder (Supplementary Fig. 15).”

Supplementary Note 1

“We observed that PtC₆₀ exhibits improved hydrophilicity compared with the Pt NCs. Specifically, the contact angles of the C₆₀ bulk crystal, Pt NCs and PtC₆₀ were determined to be 112°, 58° and 24°, respectively, as shown in Supplementary Fig. 24a-b. As a result, we believe the enhanced hydrophilicity of PtC₆₀, which inhibits the

formation of large H₂ bubbles, can prevent the structural damage to the catalyst layer caused by bubble formation,^{41–43} and further contribute to its long-term stability for HER.”

Reference

1. Yan, X., Jia, Y., Chen, J., Zhu, Z. & Yao, X. Defective-Activated-Carbon-Supported Mn–Co Nanoparticles as a Highly Efficient Electrocatalyst for Oxygen Reduction. *Adv. Mater.* **28**, 8771–8778 (2016).
2. Kessler, F. K. *et al.* Functional carbon nitride materials — design strategies for electrochemical devices. *Nat. Rev. Mater.* **2**, 17030 (2017).
3. Zhuang, Z. *et al.* Nickel supported on nitrogen-doped carbon nanotubes as hydrogen oxidation reaction catalyst in alkaline electrolyte. *Nat. Commun.* **7**, 10141 (2016).
4. S., S. N., L., S., J., H. A. & F., W. Photoinduced Electron Transfer from a Conducting Polymer to Buckminsterfullerene. *Science (80-.)*. **258**, 1474–1476 (1992).
5. Rao, A. M. *et al.* Photoinduced Polymerization of Solid C60 Films. *Science (80-.)*. **259**, 955–957 (1993).
6. Blank, V. D. *et al.* High-pressure polymerized phases of C60. *Carbon N. Y.* **36**, 319–343 (1998).
7. Gao, R., Dai, Q., Du, F., Yan, D. & Dai, L. C60-Adsorbed Single-Walled Carbon Nanotubes as Metal-Free, pH-Universal, and Multifunctional Catalysts for Oxygen Reduction, Oxygen Evolution, and Hydrogen Evolution. *J. Am. Chem. Soc.* **141**, 11658–11666 (2019).
8. Ahsan, M. A. *et al.* Tuning the Intermolecular Electron Transfer of Low-Dimensional and Metal-Free BCN/C60 Electrocatalysts via Interfacial Defects for Efficient Hydrogen and Oxygen Electrochemistry. *J. Am. Chem. Soc.* **143**, 1203–1215 (2021).
9. Miyazawa, K. *et al.* Structural characterization of C60 nanowhiskers formed by the liquid/liquid interfacial precipitation method. *Surf. Interface Anal.* **35**, 117–120 (2003).
10. Fernandez-Delgado, O. *et al.* Facile synthesis of C60-nano materials and their application in high-performance water splitting electrocatalysis. *Sustain. Energy Fuels* **4**, 2900–2906 (2020).
11. Wang, L.-L. & Cheng, H.-P. Rotation, translation, charge transfer, and electronic structure of $\{\text{C}\}_{60}$ on Cu(111) surface. *Phys. Rev. B* **69**, 45404 (2004).
12. Oh, N. K. *et al.* Highly efficient and robust noble-metal free bifunctional water electrolysis catalyst achieved via complementary charge transfer. *Nat. Commun.* **12**, 4606 (2021).
13. Oh, N. K. *et al.* In-situ local phase-transitioned MoSe₂ in La_{0.5}Sr_{0.5}CoO_{3-δ}

- heterostructure and stable overall water electrolysis over 1000 hours. *Nat. Commun.* **10**, 1723 (2019).
14. Lamoureux, P. S., Singh, A. R. & Chan, K. pH Effects on Hydrogen Evolution and Oxidation over Pt(111): Insights from First-Principles. *ACS Catal.* **9**, 6194–6201 (2019).
 15. Wang, J. *et al.* Manipulating the Water Dissociation Electrocatalytic Sites of Bimetallic Nickel-Based Alloys for Highly Efficient Alkaline Hydrogen Evolution. *Angew. Chemie Int. Ed.* **61**, e202202518 (2022).
 16. van Spronsen, M. A., Frenken, J. W. M. & Groot, I. M. N. Observing the oxidation of platinum. *Nat. Commun.* **8**, 429 (2017).
 17. Yin, H. *et al.* Ultrathin platinum nanowires grown on single-layered nickel hydroxide with high hydrogen evolution activity. *Nat. Commun.* **6**, 6430 (2015).
 18. Li, P. *et al.* Nickel single atom-decorated carbon nanosheets as multifunctional electrocatalyst supports toward efficient alkaline hydrogen evolution. *Nano Energy* **83**, 105850 (2021).
 19. Fung, S. C. XPS studies of strong metal-support interactions (SMSI)—PtTiO₂. *J. Catal.* **76**, 225–230 (1982).
 20. Liu, J. *et al.* Metal-free efficient photocatalyst for stable visible water splitting via a two-electron pathway. *Science (80-.)*. **347**, 970–974 (2015).
 21. Lee, K., Kim, S. W., Toda, Y., Matsuishi, S. & Hosono, H. Dicalcium nitride as a two-dimensional electride with an anionic electron layer. *Nature* **494**, 336–340 (2013).
 22. Jie, Z., Wenchao, S., Zhongbin, Z., Bingjun, X. & Yushan, Y. Universal dependence of hydrogen oxidation and evolution reaction activity of platinum-group metals on pH and hydrogen binding energy. *Sci. Adv.* **2**, e1501602 (2022).
 23. Rudi, S., Cui, C., Gan, L. & Strasser, P. Comparative Study of the Electrocatalytically Active Surface Areas (ECSAs) of Pt Alloy Nanoparticles Evaluated by Hupd and CO-stripping voltammetry. *Electrocatalysis* **5**, 408–418 (2014).
 24. Krivina, R. A. *et al.* Anode Catalysts in Anion-Exchange-Membrane Electrolysis without Supporting Electrolyte: Conductivity, Dynamics, and Ionomer Degradation. *Adv. Mater.* **34**, 2203033 (2022).
 25. Hodges, A. *et al.* A high-performance capillary-fed electrolysis cell promises more cost-competitive renewable hydrogen. *Nat. Commun.* **13**, 1304 (2022).
 26. Mori, M., Mržljak, T., Drobnič, B. & Sekavčnik, M. Integral Characteristics of Hydrogen Production in Alkaline Electrolysers. *Strojniški Vestn. - J. Mech. Eng.* **59**, 585–594 (2013).
 27. King, L. A. *et al.* A non-precious metal hydrogen catalyst in a commercial polymer electrolyte membrane electrolyser. *Nat. Nanotechnol.* **14**, 1071–1074 (2019).
 28. Li, D. *et al.* Highly quaternized polystyrene ionomers for high performance anion exchange membrane water electrolysers. *Nat. Energy* **5**, 378–385 (2020).
 29. Yang, Y. *et al.* Anion-exchange membrane water electrolysers and fuel cells. *Chem. Soc. Rev.* **51**, 9620–9693 (2022).

30. Vincent, I. & Bessarabov, D. Low cost hydrogen production by anion exchange membrane electrolysis: A review. *Renew. Sustain. Energy Rev.* **81**, 1690–1704 (2018).
31. Saba, S. M., Müller, M., Robinius, M. & Stolten, D. The investment costs of electrolysis – A comparison of cost studies from the past 30 years. *Int. J. Hydrogen Energy* **43**, 1209–1223 (2018).
32. Pavel, C. C. *et al.* Highly Efficient Platinum Group Metal Free Based Membrane-Electrode Assembly for Anion Exchange Membrane Water Electrolysis. *Angew. Chemie Int. Ed.* **53**, 1378–1381 (2014).
33. Mohammed-Ibrahim, J. & Sun, X. Recent progress on earth abundant electrocatalysts for hydrogen evolution reaction (HER) in alkaline medium to achieve efficient water splitting – A review. *J. Energy Chem.* **34**, 111–160 (2019).
34. Sathish, M. & Miyazawa, K. Size-Tunable Hexagonal Fullerene (C60) Nanosheets at the Liquid–Liquid Interface. *J. Am. Chem. Soc.* **129**, 13816–13817 (2007).
35. Hou, L. *et al.* Synthesis of a monolayer fullerene network. *Nature* **606**, 507–510 (2022).
36. Iwasa, Y. *et al.* New Phases of C60 Synthesized at High Pressure. *Science (80-.)*. **264**, 1570–1572 (1994).
37. Venkateswaran, U. D. *et al.* Optical Properties of Pressure-Polymerized C60. *Phys. status solidi* **198**, 545–552 (1996).
38. Zheng, J., Yan, Y. & Xu, B. Correcting the Hydrogen Diffusion Limitation in Rotating Disk Electrode Measurements of Hydrogen Evolution Reaction Kinetics. *J. Electrochem. Soc.* **162**, F1470–F1481 (2015).
39. Hansen, J. N. *et al.* Is There Anything Better than Pt for HER? *ACS Energy Lett.* **6**, 1175–1180 (2021).
40. Holton, O. T. & Stevenson, J. W. The role of platinum in proton exchange membrane fuel cells. *Platin. Met. Rev.* **57**, 259–271 (2013).
41. Xu, W., Lu, Z., Sun, X., Jiang, L. & Duan, X. Superwetting Electrodes for Gas-Involving Electrocatalysis. *Acc. Chem. Res.* **51**, 1590–1598 (2018).
42. Chahine, G. L., Kapahi, A., Choi, J.-K. & Hsiao, C.-T. Modeling of surface cleaning by cavitation bubble dynamics and collapse. *Ultrason. Sonochem.* **29**, 528–549 (2016).
43. Teran, L. A., Laín, S., Jung, S. & Rodríguez, S. A. Surface damage caused by the interaction of particles and a spark-generated bubble near a solid wall. *Wear* **438–439**, 203076 (2019).

REVIEWERS' COMMENTS

Reviewer #1 (Remarks to the Author):

I am satisfied with the revision. I think it is suitable to be accepted.

Reviewer #2 (Remarks to the Author):

The authors have addressed my concerns fully so I think the current version could be published.